# On the Emergence of Weak-to-Strong Generalization: A Bias-Variance Perspective

## Abstract

Weak-to-strong generalization (W2SG) refers to the phenomenon where a strong student model, trained on a dataset labeled by a weak teacher, ultimately outperforms the teacher on the target task. Recent studies attribute this performance gain to the prediction misfit between the student and teacher models. In this work, we theoretically investigate the emergence of W2SG through a generalized bias-variance decomposition of Bregman divergence. Specifically, we show that the expected population risk gap between the student and teacher is quantified by the expected misfit between the two models. While this aligns with previous results, our analysis removes several restrictive assumptions, most notably, the convexity of the student's hypothesis class, required in earlier works. Moreover, we show that W2SG is more likely to emerge when the student model approximates its posterior mean teacher, rather than mimicking an individual teacher. Using a concrete example, we demonstrate that if the student model size is sufficiently large, it can indeed converge to the posterior mean teacher in expectation. Our analysis also suggests that avoiding overfitting to the teacher's supervision and reducing the entropy of student's prediction further facilitate W2SG. In addition, we show that the reverse cross-entropy loss, unlike the standard forward cross-entropy, is less sensitive to the predictive uncertainty of the teacher. Finally, we empirically verify our theoretical insights and demonstrate that incorporating the reverse cross-entropy loss consistently improves student performance.

## 1 Introduction

Superalignment for large language models (LLMs) is emerging as a fundamental challenge in ensuring the long-term safety of modern natural language processing (NLP) systems (OpenAI, 2023). A promising approach to addressing superalignment is weak-to-strong generalization (W2SG) (Burns et al., 2024), where a low-complexity weak teacher model generates pseudo-labels to supervise a high-complexity pre-trained strong student model, enabling the student to outperform the teacher on the target task. To theoretically understand the emergence of this phenomenon, recent advances in W2SG research propose several analytical frameworks (Ildiz et al., 2025; Wu & Sahai, 2025; Somerstep et al., 2025a; Dong et al., 2025; Charikar et al., 2024; Mulgund & Pabbaraju, 2025; Yao et al., 2025b;a; Lang et al., 2024; Shin et al., 2025). Among these, misfit-based analyses, introduced by Charikar et al. (2024), provide a key insight by linking the student's performance gain directly to the "misfit error", namely the discrepancy between the student's predictions and the labels provided by the teacher. While the original misfit analysis in Charikar et al. (2024) focuses on squared loss, Mulgund & Pabbaraju (2025) extends this analysis to a broader family of Bregman divergences.

However, previous misfit-based analyses have several limitations. Firstly, (Charikar et al., 2024; Mulgund & Pabbaraju, 2025) rely on a restrictive convexity assumption regarding the student's function class, an assumption typically violated in practice, especially when all parameters are trainable. Although Mulgund & Pabbaraju (2025) partially relaxes this by considering convex combinations within the student's function class, it remains unclear whether one can entirely remove the convexity assumption from misfit-based W2SG inequalities. Secondly, while Mulgund & Pabbaraju (2025) suggests that reducing the misfit error facilitates W2SG, the misfit error simultaneously quantifies the attainable performance gain, implying that zero misfit error is in fact undesirable; the work does not explicitly discuss in what sense the student should align with the teacher. In addition, (Charikar et al., 2024; Mulgund & Pabbaraju, 2025) often assume the ground-truth labeling function lies within

the student's hypothesis class, but do not analyze why the student typically needs to be larger than the teacher, nor the implications of further increasing the student's model size. Finally, although reverse KL divergence appears as a misfit term in Mulgund & Pabbaraju (2025) and has been empirically studied in Yao et al. (2025a), a theoretical comparison between cross-entropy (CE) and reverse cross-entropy (RCE) within W2SG settings remains unexplored.

In this work, we present misfit-based W2SG inequalities by entirely removing the convexity assumption on the student's hypothesis class. Specifically, our misfit-based analysis is conducted at the level of *expected population risk* ( i.e., the risk is evaluated over both data and model parameter distributions), with loss functions based on Bregman divergences. This approach brings at least two benefits. First, it enables us to apply a generalized *bias-variance decomposition of Bregman divergences* recently developed by (Pfau, 2013; Adlam et al., 2022), rather than relying on the generalized Pythagorean theorem as in (Charikar et al., 2024; Mulgund & Pabbaraju, 2025), thereby avoiding the convexity assumption. In addition, explicitly analyzing expected population risk allows us to utilize the dependency between teacher and student models, an aspect overlooked in previous misfit analyses. As a result, our results suggest that an ideal scenario for W2SG occurs when the student's predictions align with those of its "posterior mean" teacher, where the posterior is defined by conditioning the teacher's distribution on the student model, and the mean may involve a dual expectation. Algorithmically, this insight encourages training student models using averaged labels from an ensemble of teachers. Furthermore, by specializing our results to squared loss and considering a ridge regression example in the overparameterized regime, we show that aligning the student with the teacher in training (i.e., reducing the empirical misfit) is sufficient for W2SG. Notably, the misfit term decreases as the student's model size grows, especially when the ridge parameter is small. More intriguingly, we further show that when the student is sufficiently large, it converges in expectation to its "posterior mean" teacher; in this case, the misfit term remains strictly positive, thus guaranteeing the occurrence of W2SG. Extending our analysis to the case where the Bregman divergence is the KL divergence (CE loss), we derive corresponding W2SG inequalities. We theoretically demonstrate that under CE or RCE loss, reducing the entropy of the student model's predictions favors W2SG. Additionally, we provide a comparative theoretical analysis of CE and RCE in ideal scenarios, and we also show that RCE is significantly less sensitive to the teacher's prediction confidence, an especially advantageous property when inputs are ambiguous for the teacher model.

We summarize our main contributions as follows:

- **Misfit-Gain Inequality and Sufficient Conditions for W2SG.** We derive the W2SG inequalities based on forward and reverse misfit errors without the convexity assumption, showing that the student-teacher misfit quantitatively captures the performance gain. These inequalities naturally lead to a clear sufficient condition for W2SG: the student aligns with its posterior mean teacher. Under squared loss, we further concretize this condition using an overparameterized ridge regression example, showing that increasing the student's capacity drives it closer to the posterior mean teacher, thereby guaranteeing W2SG.

- **Role of Predictive Confidence in W2SG.** We extend our W2SG inequalities to CE and RCE, showing that higher predictive confidence (i.e., lower entropy in the student's predictions) strengthens W2SG. We further prove that using RCE as a misfit measure preserves the performance gains from the reverse KL, and that using RCE as the optimization objective mitigates issues associated with high uncertainty in the teacher's predictions. These results highlight the important role of predictive confidence in influencing W2SG.

- **Empirical Validation and Algorithmic Improvements.** Our empirical study validates the theoretical sufficient conditions for W2SG, confirming the effects of aligning with posterior mean teacher and model capacity. We also observe that RCE retains strong performance under uncertain teacher labels. Motivated by these findings, we propose a method that combines CE and RCE to better leverage teacher confidence and consistently improve W2SG in practice.

**Other Related Literature.** To address the challenge of superalignment (OpenAI, 2023), a growing body of research has built upon Burns et al. (2024), which empirically investigates the properties of W2SG (Shin et al., 2025; Yang et al., 2025; Goel et al., 2025; Yao et al., 2025b), and the potential of this paradigm on other tasks (Guo et al., 2024; Yang et al., 2024) or scenarios (Pawelczyk et al., 2024; Zhou et al., 2025). Additionally, various techniques are also developed to enhance the strong model's performance in W2SG, such as (Lyu et al., 2025; Ye et al., 2025; Lang et al., 2025; Agrawal et al., 2024; Sang et al., 2024; Liu & Alahi, 2024; Cui et al., 2025; Somerstep et al., 2025b). Although

these methods show promising results, they often require additional weak models or involve complex computations, leading to significant time and space overhead. In parallel, theoretical understanding of W2SG mainly focuses on whether it occurs, i.e., under what circumstances the strong student outperforms the weak teacher.. From the perspective of a general definition of adversarial robustness, W2SG arises under appropriate data neighborhood conditions that enable weak supervision error correction (Lang et al., 2024) or sufficient overlap between easy and hard patterns that allow weak supervision to guide the student in learning challenging features (Shin et al., 2025). Under Gaussian data assumptions, the theoretical foundations of W2SG are rigorously characterized through several frameworks: model and distribution shift (Ildiz et al., 2025), transfer learning (Somerstep et al., 2025a) and intrinsic dimension (Dong et al., 2025). Further theoretical insights are established through representation analysis (Xue et al., 2025), feature learning (Wu & Sahai, 2025; Oh et al., 2025; Moniri & Hassani, 2025) and random feature model (Medvedev et al., 2025).

## 2 PRELIMINARIES

**Notations.** Throughout this paper, unless otherwise stated, capital letters (e.g., $X$) denote random variables, while the corresponding lowercase letters (e.g., $x$) denote their realizations. Let $P_X$ be the distribution of $X$ and $P_{X|Y}$ be the conditional distribution of $X$ given $Y$. Conditioning on a specific realization is denoted by $P_{X|Y=y}$ or simply $P_{X|y}$. Expectations are expressed as $\mathbb{E}_X[\cdot]$. Similarly, $\mathbb{E}_{X|y}[\cdot]$ or $\mathbb{E}_X[\cdot \mid y]$ denotes expectation over $X \sim P_{X|Y=y}$.

**Weak-to-Strong Generalization (W2SG) Setup.** Let $\mathcal{X}$ and $\mathcal{Y}$ denote the instance and label domains, respectively. In the weak-to-strong (W2S) setting, we consider two hypotheses classes $\mathcal{W} \subseteq \mathbb{R}^{d_w}$ and $\mathcal{W}' \subseteq \mathbb{R}^{d_s}$, with dimensions satisfying $d_s \geq d_w$. These hypothesis classes induce corresponding function classes: the weak model class $\mathcal{F} = \{f_w : \mathcal{X} \to \mathcal{Y} \mid w \in \mathcal{W}\}$, and the strong model class $\mathcal{F}' = \{f_{w'} : \mathcal{X} \to \mathcal{Y} \mid w' \in \mathcal{W}'\}$. We assume the existence of a ground-truth labeling function $g : \mathcal{X} \to \mathcal{Y}$. Suppose we have a pre-trained, high-capacity strong model $f_{w'_0} \in \mathcal{F}'$ and a weak model $f_w \in \mathcal{F}$, which may or may not be pre-trained. The weak model is trained on a dataset $S = \{(X_j, Y_j)\}_{j=1}^m$, drawn i.i.d. from an unknown distribution $\mu$. To leverage the weak model's supervision, we generate a pseudo-labeled dataset $S' = \{(X'_i, Y'_i)\}_{i=1}^n$, where $\{X'_i\}_{i=1}^n$ are drawn i.i.d. from $\mu_{\mathcal{X}}$ (which is the marginal distribution of $X$ induced by $\mu$) and each pseudo-label is obtained as $Y'_i = f_w(X'_i)$. Using this dataset, we fine-tune the pre-trained strong model $f_{w'_0}$ to obtain a final strong model $f_{w'}$. We say that W2SG occurs if this final strong model $f_{w'}$ estimates the ground-truth labeling function $g$ better than its weak teacher $f_w$.

**Remark 2.1** (Posterior Distribution of $f_W$). *The joint distribution of the teacher and student models, $P_{W,W'}$, is determined by: the marginal distribution of the teacher, $P_W$, which depends on the data distribution and other training randomness; and the conditional distribution $P_{W'|W}$, which captures the student's training given the teacher, including the influence of $S'$ and additional randomness such as pre-trained parameters $W_0$. Consequently, the conditional distribution $P_{W|W'}$ is well-defined and referred to as the "posterior" distribution of the teacher.*

To formally quantify how well a model estimates the ground-truth labeling function, a suitable loss function is typically employed. We now introduce the Bregman divergence (Bregman, 1967), a general class of divergences containing many popular loss functions.

**Definition 2.1.** Let $\phi : \mathbb{R}^d \to \mathbb{R}$ be a differentiable, strictly convex function. The Bregman divergence generated by $\phi$ of $x, y \in \mathbb{R}^d$ is defined as $\mathrm{D}_\phi(x, y) \triangleq \phi(x) - \phi(y) - \langle \nabla\phi(y), x - y \rangle$.

Bregman divergences measure the difference between a function and its linear approximation at a point, which are not true metrics, as they lack symmetry and do not satisfy the triangle inequality. Instead, they obey a generalized law of cosines (Chen & Teboulle, 1993). Common examples include the mean squared error and the KL divergence, both of which are used in machine learning.

Let $\phi^*$ be the convex conjugate[1] of $\phi$, and $x^* = \nabla\phi(x)$ be the dual representation of $x$ under $\phi$ (we have $x = (x^*)^* = \nabla\phi^*(\nabla\phi(x))$ by properties of convex conjugation). Then, the generalized law of cosines for the Bregman divergence states that, for any $x, y, z \in \mathbb{R}^d$,

$$\mathrm{D}_\phi(x, z) = \mathrm{D}_\phi(x, y) + \mathrm{D}_\phi(y, z) - \langle z^* - y^*, x - y \rangle. \tag{1}$$

---

[1] For a function $\phi : \mathcal{X} \to \mathbb{R} \cup \{-\infty, +\infty\}$, its convex conjugate is $\phi^*(y) \triangleq \sup_{x \in \mathrm{dom}(\phi)} \langle x, y \rangle - \phi(x)$.

Based on these properties, we formally define the dual expectation as follows.

**Definition 2.2** (Dual Expectation). The *dual expectation* of a random variable $X \in \mathbb{R}^d$ is defined as

$$\mathcal{E}[X] = (\mathbb{E}[X^*])^* = \nabla \phi^*(\mathbb{E}[\nabla \phi(X)]).$$

Below are two important minimization properties of Bregman divergences derived from Eq. (1).

**Lemma 2.1** (Pfau (2013); Adlam et al. (2022)). *Let $X$ be a random variable over $\mathbb{R}^d$, and $\mathrm{D}_\phi$ be any Bregman divergence over $\mathbb{R}^d \times \mathbb{R}^d$. The minimizers of the expected Bregman divergence satisfy: (i) $\arg\min_{y \in \mathbb{R}^d} \mathbb{E}[\mathrm{D}_\phi(X, y)] = \mathbb{E}[X]$. (ii) $\arg\min_{y \in \mathbb{R}^d} \mathbb{E}[\mathrm{D}_\phi(y, X)] = \mathcal{E}[X]$.*

Following (Pfau, 2013; Adlam et al., 2022), Lemma 2.1 allows us to derive the following bias-variance decompositions for the Bregman divergence with respect to an arbitrary point $y$,

$$\mathbb{E}_X[\mathrm{D}_\phi(X, y)] = \mathbb{E}_X[\mathrm{D}_\phi(X, \mathbb{E}[X])] + \mathrm{D}_\phi(\mathbb{E}[X], y), \tag{2}$$

$$\mathbb{E}_X[\mathrm{D}_\phi(y, X)] = \mathrm{D}_\phi(y, \mathcal{E}[X]) + \mathbb{E}_X[\mathrm{D}_\phi(\mathcal{E}[X], X)]. \tag{3}$$

Given a Bregman divergence as the loss function, we formally define the *expected population risk* of the teacher model $f_W$ as $\mathbb{E}_{X,W}[\mathrm{D}_\phi(g(X), f_W(X))]$, where $X$ denotes an independent test sample. Since the Bregman divergence is generally asymmetric, we also define the corresponding *reverse population risk* as $\mathbb{E}_{X,W}[\mathrm{D}_\phi(f_W(X), g(X))]$. The population risks for the student model are defined analogously. Both the population risk and reverse population risk serve as measures of how well a model approximates the ground-truth labeling function $g$.

We next restate the previous misfit-based W2SG inequality from Mulgund & Pabbaraju (2025).

**Theorem 2.1** (Informal, Bregman Misfit-Gain Inequality in Mulgund & Pabbaraju (2025)). *Let $f_{w'}$ and $f_w$ be the strong and weak models, obtained by adding a finetuning layer on top of their respective pre-trained backbones. The student's finetuning function is drawn from a **convex set** $\mathcal{F}$. If $f_{w'}$ is sufficiently close to the projection of the weak model onto the strong model class, then*

$$\mathbb{E}_X[D_\phi(g(X), f_{w'}(X))] \leq \mathbb{E}_X[D_\phi(g(X), f_w(X))] - \mathbb{E}_X[D_\phi(f_{w'}(X), f_w(X))] + \epsilon.$$

*The residual error $\epsilon$ vanishes when the student model $f_{w'}$ is exactly the projection.*

The formal statement is in Appendix A. This theorem suggests that a student model can provably outperform a weak teacher, as quantified by their misfit, if the student's hypothesis class is a **convex set**. The generalized Pythagorean inequality, which is central to the proof, interprets fine-tuning the strong model with weak labels as a projection of the weak model onto the student's model class. However, this key convexity assumption does not hold in deep classification settings, since the softmax function used to normalize probabilities breaks convexity (Mulgund & Pabbaraju, 2025).

## 3 W2SG THROUGH THE LENS OF EXPECTED MISFIT

We are now in a position to present our main results to overcome the convexity assumption above.

**Theorem 3.1.** *Under the teacher-student setting, the following inequalities hold,*

$$\mathbb{E}[\mathrm{D}_\phi(g(X), f_{W'}(X))] \leq \mathbb{E}[\mathrm{D}_\phi(g(X), f_W(X))] - \mathbb{E}[\mathrm{D}_\phi(f_{W'}(X), f_W(X))] + \epsilon_1, \tag{4}$$

$$\mathbb{E}[\mathrm{D}_\phi(f_{W'}(X), g(X))] \leq \mathbb{E}[\mathrm{D}_\phi(f_W(X), g(X))] - \mathbb{E}[\mathrm{D}_\phi(f_W(X), f_{W'}(X))] + \epsilon_2, \tag{5}$$

*where $\epsilon_1 = \sqrt{\mathbb{E}\|f_{W'}^*(X) - \mathbb{E}[f_W^*(X) \mid W', X]\|^2} \sqrt{\mathbb{E}\|g(X) - f_{W'}(X)\|^2}$ and $\epsilon_2 = \sqrt{\mathbb{E}\|f_{W'}(X) - \mathbb{E}[f_W(X) \mid W', X]\|^2} \sqrt{\mathbb{E}\|g^*(X) - f_{W'}^*(X)\|^2}$.*

Eq. (4) implies that the student can outperform the teacher (i.e., W2SG occurs) if $\epsilon_1$ is sufficiently small. The potential performance gain is characterized by the *expected reverse misfit*, namely $\mathbb{E}[\mathrm{D}_\phi(f_{W'}(X), f_W(X))]$, between the two models. Conceptually, Eq. (4) aligns with Theorem 2.1 (Mulgund & Pabbaraju, 2025)(restated formally as Theorem A.2 in Appendix). However, our analysis avoids many of their assumptions, including realizability, convexity, and sequential consistency, with relaxing convexity being particularly significant. This is made possible by taking

expectations jointly over the data and model parameter distributions (i.e., $P_{X,W,W'}$)$^2$ and by invoking the Bregman divergence decomposition in Eq. (3), rather than relying on the projection of the teacher model onto a convex hypothesis set of the student. Moreover, explicitly analyzing the expected population risk allows us to capture the statistical dependency between teacher and student models, which previous misfit analyses overlook. This expectation-level formulation can also be translated into high-probability bounds using standard concentration arguments.

In addition, Eq. (5) conveys an analogous message, but focuses on reverse population risks, a direction of W2SG inequality not established by Mulgund & Pabbaraju (2025). Note that the *expected forward misfit*, namely $\mathbb{E}\left[D_\phi\left(f_W(X), f_{W'}(X)\right)\right]$, in Eq. (5), when instantiated as cross-entropy (a surrogate for KL divergence), is the standard training loss in practical W2SG setups. We will elaborate on this in Section 5.

Notably, Theorem 3.1 reveals a subtle trade-off in aligning the student model with the teacher. On one hand, since the student lacks access to the ground-truth labels, it must rely on the pseudo labels provided by the teacher. That is, minimizing the empirical risk with respect to the teacher's outputs, e.g., $\frac{1}{n}\sum_{i=1}^{n} D_\phi\left(f_w(x_i), f_{w'}(x_i)\right)$, becomes a necessary part of training. On the other hand, Eq. (4-5) suggest that the expected misfit between the teacher and the student models contributes directly to the performance improvement of the strong student over the weak teacher. In particular, greater misfit between the two models can indicate a larger performance gain achieved by the student, which also align with a recent argument given in Dong et al. (2025). This suggests that a pointwise alignment of $f_{w'}$ with $f_w$ may in fact be undesirable, especially when abundant training data is available, so regularization strategies such as early stopping may be necessary to prevent the student from fully mimicking the teacher.

Theorem 3.1 also hints conditions under which $\epsilon_1$ and $\epsilon_2$ vanish, leading to the following results.

**Corollary 3.1.** *Under the teacher-student setting, if $f_{W'}(x) = \mathcal{E}\left[f_W(x)|W'\right]$ for $\forall x \in \mathcal{X}$, then*

$$\mathbb{E}\left[D_\phi\left(g(X), f_{W'}(X)\right)\right] = \mathbb{E}\left[D_\phi\left(g(X), f_W(X)\right)\right] - \mathbb{E}\left[D_\phi\left(\mathcal{E}\left[f_W(X)|W', X\right], f_W(X)\right)\right]. \quad (6)$$

*Furthermore, if $f_{W'}(x) = \mathbb{E}\left[f_W(x)|W'\right]$ for $\forall x \in \mathcal{X}$, then*

$$\mathbb{E}\left[D_\phi\left(f_{W'}(X), g(X)\right)\right] = \mathbb{E}\left[D_\phi\left(f_W(X), g(X)\right)\right] - \mathbb{E}\left[D_\phi\left(f_W(X), \mathbb{E}\left[f_W(X)|W', X\right]\right)\right]. \quad (7)$$

Corollary 3.1 presents situations where W2SG is guaranteed to emerge: specifically, when the student's prediction matches that of its (dual) "posterior mean" teacher, where the (dual) expectation is taken with respect to $P_{W|W'}$ (i.e. the posterior distribution of teacher model, as discussed in Remark 2.1). Moreover, it is worth noting that both the forward and reverse misfit terms attain their minimum values in Eq. (6) and Eq. (7), as indicated by Lemma 2.1.

Corollary 3.1 also motivates a closer look at the regime where the student approaches its posterior mean teacher, since the relationship between the residual terms ($\epsilon_1$ and $\epsilon_2$) and the expected misfit is not yet well understood at this stage. In the next section, we demonstrate that reducing the expected misfit is sufficient to drive the residual term to zero, and that in overparameterized settings, enlarging the student can in fact enable convergence to its posterior mean teacher.

## 4 SYMMETRIC BREGMAN DIVERGENCE: SQUARED LOSS

In Mulgund & Pabbaraju (2025), the emergence of W2SG is attributed to the student being close to the convex projection of the teacher, but it is unclear why fine-tuning should lead to this projection. In contrast, our Theorem 3.1 suggests that W2SG arises when the student model approximates the posterior mean teacher. We now investigate the conditions under which the student model becomes close to its posterior mean teacher. Specifically, in this section, we focus on the squared loss (i.e. $\phi(x) = \|x\|^2$), where $\epsilon_1 = \epsilon_2$. The following result is a special case of Theorem 3.1.

**Corollary 4.1.** *Let $\phi(x) = \|x\|^2$, then the following inequality holds,*

$$\mathbb{E}\|g(X) - f_{W'}(X)\|^2 \leq \mathbb{E}\|g(X) - f_W(X)\|^2 - \mathbb{E}\|f_{W'}(X) - f_W(X)\|^2 + \epsilon_2,$$

*where $\epsilon_2 = 2\sqrt{\mathbb{E}\left\|f_{W'}(X) - \mathbb{E}\left[f_W(X) \mid W', X\right]\right\|^2}\sqrt{\mathbb{E}\|g(X) - f_{W'}(X)\|^2}$.*

---

$^2$To avoid clutter, expectation subscripts are omitted in theorem statements.

To analyze when $\epsilon_2$ becomes small, we focus on the quantity $\mathbb{E}\left\|f_{W'}(X) - \mathbb{E}\left[f_W(X) \mid W', X\right]\right\|^2$, rather than directly analyzing the expected distance between the student model and ground truth labeling function, i.e. $\mathbb{E}\left\|g(X) - f_{W'}(X)\right\|^2$, which is also a component of $\epsilon_2$. For any input $x \in \mathcal{X}$,

$$\mathbb{E}\left\|f_{W'}(x) - \mathbb{E}\left[f_W(x) \mid W'\right]\right\|^2 = \underbrace{\mathbb{E}\left\|f_{W'}(x) - f_W(x)\right\|^2}_{\text{Expected Misfit}} - \underbrace{\mathbb{E}\left\|f_W(x) - \mathbb{E}\left[f_W(x) \mid W'\right]\right\|^2}_{\text{Conditional Variance of } f_W}. \quad (8)$$

Notably, due to the presence of the conditional variance, we can see that $\epsilon_2$ vanishes before the expected misfit does, in other words, the vanishing of the expected misfit term is a sufficient condition for $\epsilon_2$ to vanish. Furthermore, recent studies on the "double descent" phenomenon show that this expected misfit, when regarded as the population risk, can indeed decrease as the capacity of the student model increases (Belkin et al., 2019; Hastie et al., 2022; Mei & Montanari, 2022; Yang et al., 2020; Ba et al., 2020). We now formalize this through the following example.

**Example 1** (Ridge Regression). Assume $X \sim \mathcal{N}\left(0, \mathbf{I}_{d_w}/d_w\right)$, and let the teacher model be $f_w(x) = x^\top w$ for $w \in \mathbb{R}^{d_w}$ and the student model be $f_{w'}(x) = (w'_1 x)^\top w'_2$, where[3] $w'_1 \in \mathbb{R}^{d_s \times d_w}$ and $w'_2 \in \mathbb{R}^{d_s}$. The weights $W'_1$ are initialized with entries drawn i.i.d. from $\mathcal{N}(0, 1/d_w)$ and remain fixed during training. The student is trained via ridge regression: $\min_{w'_2}\left\|(w'_1 \mathbf{x}')^\top w'_2 - \mathbf{y}'\right\|^2 + \eta\|w'_2\|^2$ where $\mathbf{x}' = [x'_1, \ldots, x'_n] \in \mathbb{R}^{d_w \times n}$ and $\mathbf{y}' = [y'_1, \ldots, y'_n] \in \mathbb{R}^n$.

Consider an overparameterized setting, we derive the following asymptotic result.

**Theorem 4.1.** *Assume* $W \sim \mathcal{N}(\mu, B\mathbf{I}_{d_w})$ *where* $\mu \in \mathbb{R}^{d_w}$ *is independent of* $W'_1$ *and* $\mathbf{X}'$ *and* $\|\mu\|^2 \leq C$ *for some fixed constant* $C$. *Suppose there exist* $\gamma_1 \in (1, \infty)$ *and* $\gamma_2 \in (0, 1)$ *s.t.* $\frac{d_s}{d_w} \to \gamma_1$ *and* $\frac{n}{d_w} \to \gamma_2$ *as* $n, d_w, d_s \to \infty$. *Then, as* $n, d_s, d_w \to \infty$, *we have*

$$\mathbb{E}\|f_{W'}(X) - f_W(X)\|^2 = Bh(\eta, \gamma_1, \gamma_2),$$

*where* $h(\eta, \gamma_1, \gamma_2) = 1 - \gamma_2 + \gamma_2 \kappa^2 (\kappa + \gamma_1 - 1) \frac{1 - \kappa - \gamma_1}{(1-\kappa)^2 - \gamma_1}$ *and* $\kappa \in (0, 1)$ *is the unique solution to* $\eta^2(1 - \kappa) = \gamma_2 \kappa (\kappa + \gamma_1 - 1)(1 - \gamma_2 + \gamma_2 \kappa)$.

As in many random matrix theory–based analyses of double descent, the behavior of the function $h(\eta, \gamma_1, \gamma_2)$ is not immediately transparent from its expression. To gain insight, we visualize it numerically (see Figure 3 in the Appendix). In fact, for small $\eta$ (i.e. when regularization is not too strong), $h(\eta, \gamma_1, \gamma_2)$ decreases almost monotonically as $\gamma_1$ increases, highlighting the benefit of enlarging the student model since a smaller misfit error reduces $\epsilon_2$. Furthermore, regardless of the value of $\eta$, the following corollary identifies a setting in which the misfit term remains nonzero while $\epsilon_2$ vanishes, namely a regime where W2SG is guaranteed to occur.

**Corollary 4.2.** *Under the setting of Theorem 4.1, if* $\gamma_1 \to \infty$, *we have* $\epsilon_2 \to 0$ *and* $\mathbb{E}\|f_{W'}(X) - f_W(X)\|^2 \to B(1 - \gamma_2)$.

**Remark 4.1.** *Corollary 4.2 shows that when the student model is sufficiently large, the misfit term converges to the conditional variance of* $f_W$ *(i.e. the RHS of Eq. (8) vanishes). In this regime,* $\epsilon_2 \to 0$ *while* $\mathbb{E}\|f_{W'}(X) - f_W(X)\|^2 \neq 0$. *Consequently, the student converges in expectation to its posterior mean teacher, yet a nonzero performance gain remains, namely W2SG emerges. Importantly, this implies that enlarging the student promotes the occurrence of W2SG but does not necessarily lead to the optimal performance gain.*

Example 1 and Theorem 4.1 can be extended to nonlinear neural networks by utilizing the analysis in Mei & Montanari (2022). In summary, W2SG is more likely to arise when the student model is sufficiently large (i.e., large $\gamma_1$), but the performance gain eventually saturates; as the student size grows, the performance gain converges to its minimum value (e.g., $B(1 - \gamma_2)$ in Example 1). These observations have not been explicitly highlighted in previous misfit-based analyses (Charikar et al., 2024; Mulgund & Pabbaraju, 2025), which typically assume that the ground-truth labeling function lies within the student's hypothesis class without discussing why a larger student model is necessary.

While we show that enlarging the student model enables convergence to its posterior mean teacher under a symmetric Bregman divergence loss (specifically, squared loss), we conjecture that a similar result should extend to asymmetric Bregman divergences. Establishing this theoretically, however, remains challenging. A key obstacle is the absence of formal analyses of double descent behavior under cross-entropy loss, despite its well-documented empirical evidence (Nakkiran et al., 2020).

---

[3] Here $d_s$ refers to the number of hidden units rather than the total number of parameters.

## 5 ASYMMETRIC BREGMAN DIVERGENCE: FROM KL TO CE

Although providing a formal justification for the vanishing of $\epsilon_1$ or $\epsilon_2$ under asymmetric Bregman divergences is difficult, our two-direction misfit-based W2SG inequalities in Theorem 3.1 nevertheless provide valuable insights into this setting. We now turn to a $K$-class classification task, where $\mathcal{Y} \subset \mathbb{R}^K$ and $\|y\|_1 = 1$ for all $y \in \mathcal{Y}$, and consider the commonly used CE loss function, defined as $\mathrm{CE}(y, \hat{y}) \triangleq -\sum_{i=1}^K y_i \log \hat{y}_i$ for any $y, \hat{y} \in \mathcal{Y}$. Since $\mathrm{CE}(y, \hat{y}) = \mathrm{D_{KL}}(y\|\hat{y}) + H(y)$, where $\mathrm{D_{KL}}(y\|\hat{y}) \triangleq \sum_{i=1}^K y_i \log \frac{y_i}{\hat{y}_i}$ is the KL divergence and $H(y) = -\sum_{i=1}^K y_i \log y_i$ is the Shannon entropy, and given that KL divergence is a special case of Bregman divergence, we have:

**Corollary 5.1.** *Let $\phi(x) = \sum x_i \log x_i$ and define the reverse cross-entropy (RCE) as $\mathrm{RCE}(y, \hat{y}) \triangleq -\sum_{i=1}^K \hat{y}_i \log y_i$ for any $y, \hat{y} \in \mathcal{Y}$, the following inequalities hold,*

$$\mathbb{E}\left[\mathrm{CE}(g(X), f_{W'}(X))\right] \leq \mathbb{E}\left[\mathrm{CE}(g(X), f_W(X))\right] - \mathbb{E}\left[\mathrm{RCE}(f_W(X), f_{W'}(X))\right] + \mathbb{E}\left[H(f_{W'}(X))\right] + \epsilon_1,$$
$$\mathbb{E}\left[\mathrm{RCE}(g(X), f_{W'}(X))\right] \leq \mathbb{E}\left[\mathrm{RCE}(g(X), f_W(X))\right] - \mathbb{E}\left[\mathrm{CE}(f_W(X), f_{W'}(X))\right] + \mathbb{E}\left[H(f_{W'}(X))\right] + \epsilon_2,$$

*where $\epsilon_1, \epsilon_2$ are defined as in Theorem 3.1.*

Notably, both inequalities in Corollary 5.1 involve the entropy of the student's prediction, $H(f_{W'}(X))$. This implies that, when CE or RCE is used to measure population risk, reducing the entropy of the student's output distribution favors the emergence of W2SG. In other words, high-confidence predictions by the student (i.e., low predictive uncertainty) are beneficial for outperforming the teacher. In fact, the original W2SG paper (Burns et al., 2024) adopts a regularized loss of the form $\mathcal{L}_{\mathrm{AUX}} = \beta \mathrm{CE}(f_w(x), f_{w'}(x)) + (1-\beta)\mathrm{CE}\left(\hat{f}_{w'}(x), f_{w'}(x)\right)$, where $\hat{f}_{w'}(x)$ is the hardened student 's prediction so $\hat{f}_{w'}(x)$ is a one-hot vector. This regularization explicitly encourages entropy minimization, consistent with the implications of Corollary 5.1.

Furthermore, evaluating the quality of a trained model using the forward CE is the de facto standard in practice, then according to Corollary 5.1, minimizing RCE between the teacher and student appears more natural. As conjectured at the end of Section 4, this may help reduce $\epsilon_1$ and thereby facilitate W2SG. Empirical studies by Yao et al. (2025a) have also advocated reverse KL for its mode-seeking property, which benefits strong model performance. We now proceed to elaborate on the comparison between CE and RCE as objective functions in W2S training.

In the idealized setting where both $\epsilon_1$ and $\epsilon_2$ vanish, we obtain the following result.

**Proposition 1.** *Under the ideal conditions of Corollary 3.1 where $\epsilon_1$ and $\epsilon_2$ vanish, the performance gain in the first inequality of Corollary 5.1 coincides with Eq. (6), while the gain in the second inequality is strictly smaller than Eq. (7).*

Thus, evaluating the population risk using CE, while aligning the student to the teacher via RCE, preserves the performance gains predicted by Corollary 3.1. Moreover, RCE provides additional notable advantages when the teacher's predictions exhibit low confidence, as illustrated below.

**Proposition 2.** *Given a data distribution $\mu = \mu_X \mu_{Y|X}$ and any $\alpha \in [0, 1]$, we define a label-shifted distribution $\hat{\mu} = \mu_X \mu_{\hat{Y}|X}$ by smoothing the labels as follows: for each $(X, Y) \sim \mu$, the smoothed label is given by $\hat{Y}_j = \frac{1}{2} + \alpha\left(Y_j - \frac{1}{2}\right)$ for $\forall j \in [2]$. If RCE is used as the loss function in binary classification, then the population risk minimizer on $\hat{\mu}$ also minimizes the population risk on $\mu$.*

Note that decreasing $\alpha$ increases the uncertainty of $\hat{Y}$ but does not affect its hard label. If $\hat{Y}$ is the label provided by the teacher, Proposition 2 shows that RCE is less sensitive to the teacher's confidence levels, which is especially desirable when the input $x$ is ambiguous for the teacher. Furthermore, in Appendix D.4, we demonstrate that uncertain weak supervision can lead to vanishing gradients under standard CE, whereas RCE maintains gradient stability. Later on we will empirically demonstrate how to use the advantages of RCE in scenarios involving low-confidence labels.

## 6 EXPERIMENTS

In this section, we present empirical studies on W2SG. Specifically, we aim to verify our theoretical insights, visualize the bias and variance terms in W2SG, and compare CE and RCE as training objectives. Additionally, we propose a novel training objective to improve the performance of W2SG.

**Datasets.** Our experiments employ diverse datasets covering both standard NLP tasks and LLM reward modeling tasks. For standard NLP tasks, we utilize the SciQ (Welbl et al., 2017), Amazon Polarity (McAuley & Leskovec, 2013) and Twitter Sentiment [4] datasets, following the experimental setup of Burns et al. (2024) to transform these datasets into binary classification tasks. For reward modeling tasks, we sample subsets from CAI-Harmless (Bai et al., 2022b) and HH-RLHF (Bai et al., 2022a), with experimental settings referencing (Yang et al., 2025), to guide models toward achieving harmlessness or helpfulness objectives.

**Models.** We employ models from the GPT-2 series (Radford et al., 2019), including GPT2, GPT2-Medium, GPT2-Large, and GPT2-XL, and the Qwen series (Bai et al., 2023), including Qwen-1.8B, Qwen-7B, and Qwen-14B. We use full fine-tuning without freezing any pretrained parameters.

**Emergence of W2SG.** We empirically investigate several insights from our theoretical framework in the context of LLMs: 1) The conditions for W2SG to emerge, as outlined in Theorem 3.1 and Corollary 3.1; 2) How the strong model's capacity influences W2SG, as discussed in Theorem 4.1 and Remark 4.1. To further deepen our understanding of these phenomena in these two scenarios, we also systematically investigate how bias and variance affect the strong model's performance. For the first insight, we independently train multiple weak teachers and use a probability-based ensemble (Dietterich, 2000; Zhou, 2025) to approximate the dual expectation term in Corollary 3.1. For the second one, we progressively scale the strong model's capacity and use Algorithm 1 (Yang et al., 2020) to estimate the changes in bias and variance. The results are shown in Figure 1.

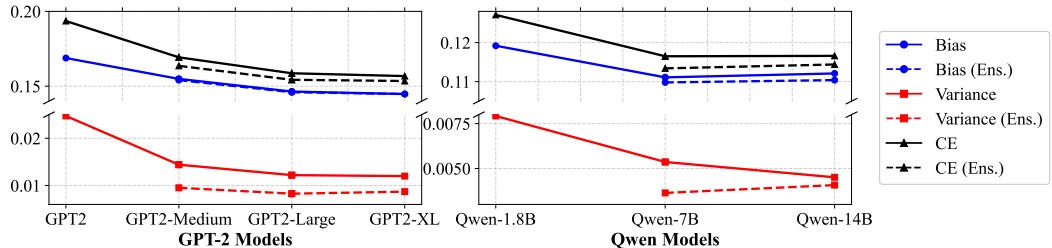

Figure 1: Bias and variance estimation on Amazon Polarity. GPT2 and Qwen-1.8B serve as weak teachers to supervise strong student models. "CE" denotes cross-entropy test loss. "Ens." denotes student performance supervised by expected teacher, approximated via weak teachers ensemble.

*Conditional expectation estimation makes W2SG emerge*. As predicted by Corollary 3.1, using model ensembles to approximate the expected teacher consistently reduces the student model's test loss compared to mimicking a single teacher, confirming W2SG is more likely when the student matches its posterior mean teacher. This aligns with prior work (Agrawal et al., 2024; Sang et al., 2024; Liu & Alahi, 2024; Cui et al., 2025) where multiple teachers train a student model. Results on other datasets, presented in Figure 4, offer further empirical support for our theory.

*Bias and variance analysis in W2SG*. Notably, scaling up student capacity primarily reduces bias. While both bias and variance decrease, the variance and its reduction remain significantly smaller than bias terms. This differs from Dong et al. (2025), which mainly focuses on variance-dominated scenarios. Moreover, these trends align with Chen et al. (2024), which shows that the trends of bias and variance are consistent, and the variance is upper bounded by the bias. Model ensembles decrease test loss mainly through variance reduction, as incorporating more weak teachers helps mitigate randomness in their outputs.

**RCE vs. CE.** To systematically investigate the roles of CE and RCE misfit in W2SG training, we compare their performance during strong model training. Specifically, we apply the label smoothing strategy in Proposition 2. A smaller smooth factor $\alpha$ leads to higher predictive uncertainty in the shifted pseudo-labels. The performance of strong model using this label smoothing strategy is shown in Figure 2. We observe that even when $\alpha = 0.001$, where the pseudo-labels are nearly uniform, RCE maintains stable performance without significant degradation. This aligns with Proposition 2. Moreover, in some cases, moderate label shifting even improves accuracy. In contrast, the performance of CE drops rapidly as $\alpha$ decreases. Additional results on larger models, standard knowledge

---

[4]https://www.kaggle.com/datasets/jp797498e/twitter-entity-sentiment-analysis

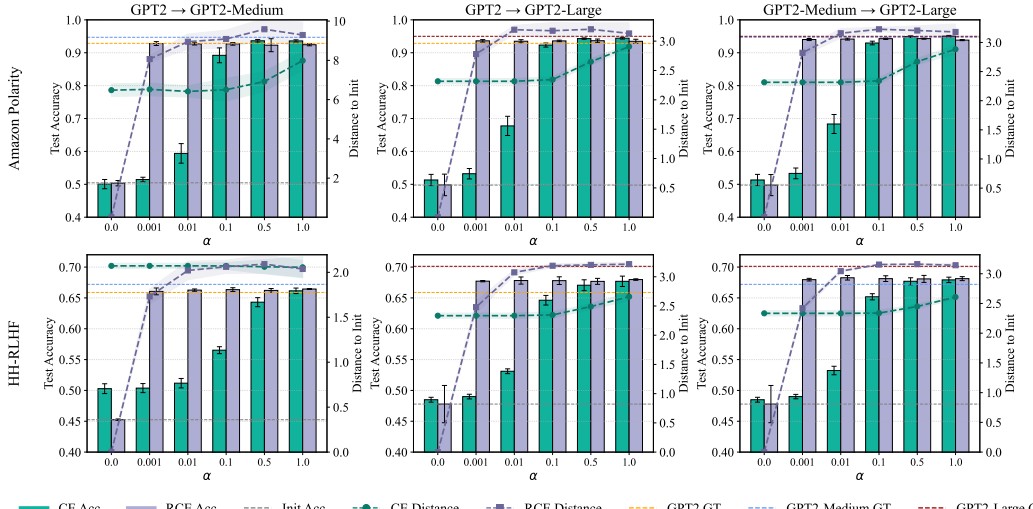

Figure 2: Performance of the GPT2 series models under varying $\alpha$ values, comparing CE and RCE losses. "GPT2 $\rightarrow$ GPT2-Medium" denotes GPT2 supervising GPT2-Medium. Left y-axis shows test accuracy. Right y-axis illustrates the $L_2$ norm between the fine-tuned and initial models, represented by the line plots for CE Distance and RCE Distance. GT denotes the test accuracy achieved by training with ground truth labels.

Table 1: Test accuracy (%) of five loss functions on multiple datasets and models. The optimal and suboptimal results are marked in **bold** and underline, respectively.

| Dataset | Model | CE | RCE | AUX | CACE | SL |
|---|---|---|---|---|---|---|
| CAI-Harmless | GPT2 $\rightarrow$ GPT2-Medium | 93.05 | 93.00 | 92.81 | **93.44** | 93.32 |
| | GPT2 $\rightarrow$ GPT2-Large | 93.88 | 94.55 | 94.51 | **94.78** | 94.63 |
| | GPT2-Medium $\rightarrow$ GPT2-Large | 95.47 | 95.40 | 95.40 | **95.62** | 95.58 |
| | Qwen-1.8B $\rightarrow$ Qwen-7B | 96.45 | 95.93 | 96.40 | **96.75** | 96.50 |
| | Qwen-1.8B $\rightarrow$ Qwen-14B | 96.03 | 94.93 | 95.65 | **96.08** | 95.70 |
| | Qwen-7B $\rightarrow$ Qwen-14B | 96.48 | 95.75 | 96.13 | **96.58** | 96.13 |
| HH-RLHF | GPT2 $\rightarrow$ GPT2-Medium | 66.15 | 66.44 | 66.21 | 66.18 | **66.65** |
| | GPT2 $\rightarrow$ GPT2-Large | 67.69 | **67.97** | 67.73 | 67.82 | **67.97** |
| | GPT2-Medium $\rightarrow$ GPT2-Large | 67.93 | 68.13 | 68.08 | 68.11 | **68.53** |
| | Qwen-1.8B $\rightarrow$ Qwen-7B | 68.45 | 69.08 | 68.80 | 69.20 | **69.25** |
| | Qwen-1.8B $\rightarrow$ Qwen-14B | 66.03 | 67.45 | 67.73 | **69.20** | 68.65 |
| | Qwen-7B $\rightarrow$ Qwen-14B | 67.90 | 69.13 | 69.20 | 69.78 | **70.85** |

distillation settings, and KL/Reverse KL objectives are provided in Appendix E.3. We also observe that under RCE supervision, the model moves farther from its initialization compared to CE (corresponding to the RCE/CE distances in Figure 2); an empirical explanation of this phenomenon is provided in Appendix E.3.

Moreover, Proposition 2 and Figure 2 indicate that RCE might be more suitable for low-confidence samples. Therefore, a combination of CE and RCE may better exploit the information within weak supervision. To explore this idea, we propose confidence-adaptive cross entropy (CACE) loss as: $\mathcal{L}_{\text{CACE}}(y, \hat{y}) = \mathbb{I}(y, c) \cdot \text{RCE}(y, \hat{y}) + (1 - \mathbb{I}(y, c)) \cdot \text{CE}(y, \hat{y})$, where $c$ is the confidence threshold, and $\mathbb{I}(y, c)$ is an indicator function that activates when the confidence of the soft label $y$ is below $c$. We note the symmetric cross entropy loss (SL) (Wang et al., 2019) shares a similar design philosophy with ours, defined as: $\mathcal{L}_{\text{SL}}(y, \hat{y}) = \lambda_1 \text{RCE}(y, \hat{y}) + \lambda_2 \text{CE}(y, \hat{y})$. Additionally, we compare our method with the auxiliary confidence loss (AUX) from Burns et al. (2024). As shown in Table 1, CACE or SL consistently outperforms other three losses. While AUX improves student confidence via regularization, it requires careful hyperparameter tuning for the warm-up phase. In contrast, CACE and SL leverage the strengths of CE and RCE based on the weak label confidence, without

relying on the capacity of student models. The results on the other datasets are presented in Table 3 in Appendix F.

## 7 CONCLUSION AND LIMITATIONS

This work provides a theoretical and empirical analysis of W2SG. We successfully remove restrictive assumptions from the previous misfit-based W2SG theory and show that a sufficient condition for W2SG is when the student approximates its posterior mean teacher, a property achievable by increasing the student's model size. We further demonstrate the effectiveness of RCE in handling uncertain supervision. Our experiments confirm that larger student models and ensemble-based supervision improve W2SG. We also find that RCE is more resilient to low-confidence pseudo-labels and that combining RCE with standard CE in a confidence-adaptive way yields better results. However, some theoretical insights are primarily validated in ridge regression and rely on Gaussian assumptions, which represents a general limitation of current theoretical approaches to W2SG. Future work could extend this framework to broader architectures and real-world applications.

## REPRODUCIBILITY STATEMENT

We are committed to ensuring the reproducibility of our work. To facilitate the verification of our theoretical contributions, we provide complete proofs for the results in Appendix B, C, and D, respectively. For our empirical results, a comprehensive description of the experimental setup, including datasets, model architectures, and training configurations, is detailed in Appendix E.

## ETHICS STATEMENT

Our work investigates weak-to-strong generalization, a method to enhance the safety and alignment of large language models. We adhere to the ICLR Code of Ethics. We acknowledge that the W2SG paradigm could potentially be applied in settings where model alignment has societal impacts, such as in safety-critical systems or AI oversight. However, this paper focuses solely on theoretical understanding and controlled experimental validation. We do not propose or evaluate real-world deployments. This paper uses only publicly available datasets and models, containing no personally identifiable information and involving no human subjects. We confirm that there are no conflicts of interest related to this work, and for full transparency, we have disclosed our use of large language models for editing in Appendix G.

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

# Content

# Appendix

## A PREVIOUS MISFIT-BASED RESULTS

**Theorem A.1** (Restatement of Theorem 1 from Charikar et al. (2024)). *This Theorem considers the squared loss as a special case of Bregman divergence $D_\phi(\cdot, \cdot)$, i.e., $\phi(x) = \|x\|^2$. Let $h_s : \mathbb{R}^d \to \mathbb{R}^{d_s}$ and $h_w : \mathbb{R}^d \to \mathbb{R}^{d_w}$ be the strong and weak model representation maps respectively. Given some data labeled by $g$, let $f_w \circ h_w$ be the function learnt by the weak model, for some classier head $f_w : \mathbb{R}^{d_w} \to \mathbb{R}$. For a convex set of functions $\mathcal{F}_s$ mapping $\mathbb{R}^{d_s}$ to $\mathbb{R}$ let*

$$f_{sw} = \mathrm{argmin}_{f \in \mathcal{F}_s} \mathbb{E}_X \|f(h_s(X)) - f_w(h_w(X))\|^2,$$

*be the function learnt by the strong model under weak supervision. Lastly, let us assume that there exists $f_s \in \mathcal{F}_s$ such that $f_s \circ h_s = g$. Then, we have that*

$$\mathbb{E}_X \|f_{sw}(h_s(X)) - g(X)\|^2 \leq$$
$$\mathbb{E}_X \|f_w(h_w(X)) - g(X)\|^2 - \mathbb{E}_X \|f_{sw}(h_s(X)) - f_w(h_w(X))\|^2. \quad (9)$$

**Remark A.1.** *In comparison, our bound in Corollary 4.1 is*

$$\mathbb{E}_{X,W'} \|g(X) - f_{W'}(X)\|^2 \leq \mathbb{E}_{X,W'} \|g(X) - f_W(X)\|^2 - \mathbb{E}_{X,W',W} \|f_{W'}(X) - f_W(X)\|^2 + \epsilon_2,$$

*Our bound differs from Theorem A.1 in that it focuses on the expected population risk, that is, the expectation is taken over both the data distribution and the model parameters. As mentioned in the main text, this enables us to remove the convexity assumption by invoking Lemma 2.1. Notably, when restricted to convex function classes, our result recovers Theorem A.1, as convexity guarantees the last term $\mathbb{E}_{W'} \langle \mathbb{E}_{W|W'}[f_W^*(x)] - f_{W'}^*(x), g(x) - f_{W'}(x) \rangle \leq 0$ in Eq. (15) in our framework, following similar developments as those in Charikar et al. (2024). This recovery can also be seen by comparing (Charikar et al., 2024, Eq. (8)) with our Eq. (15). Specifically, taking expectations over both $(W, W') \sim P_{W,W'}$ in their bound obtains a result similar to ours, corresponding to a non-positive residual term in Eq. (15) in our proof.*

**Theorem A.2** (Restatement of Theorem 4.1 from Mulgund & Pabbaraju (2025)). *Let $\phi$ be a proper convex function as defined in Definition 2.1. Let $h_s, h_w$ be defined in the same way in Theorem A.1. Let $f_w : \mathbb{R}^{d_w} \to \mathcal{Y}$ be the weak model finetune layer, and $g : \mathcal{X} \to \mathcal{Y}$ be the target function. Let $\mathcal{F}$ be a class of functions mapping $\mathbb{R}^{d_s} \to \mathcal{Y}$. If the following hold:*

1. *(Realizability) $\exists f_* \in \mathcal{F}$ s.t. $g = f_* \circ h_s$,*

2. *(Convexity) $\mathcal{F}$ is a convex set of functions,*

3. *(Sequential Consistency) For $y \in \mathcal{Y}$ fixed, if $D_\phi(x_n, y) \to 0$, then $x_n \to y$,*

*then for any $\epsilon > 0$, there exists $\delta > 0$ such that for all $f_s \in \mathcal{F}$ that satisfy*

$$\mathbb{E}_X [D_\phi(f_s(h_s(X)), f_w(h_w(X)))] \leq \inf_{f \in \mathcal{F}} \mathbb{E}_X [D_\phi(f(h_s(X)), f_w(h_w(X)))] + \delta,$$

*we have*

$$\mathbb{E}_X [D_\phi(g(X), f_s(h_s(X)))] \leq$$
$$\mathbb{E}_X [D_\phi(g(X), f_w(h_w(X)))] - \mathbb{E}_X [D_\phi(f_s(h_s(X)), f_w(h_w(X)))] + \epsilon. \quad (10)$$

**Remark A.2.** *In contrast to Theorem A.2, our bounds in Theorem 3.1 are derived without relying on any of the three assumptions mentioned earlier: realizability, convexity, and sequential consistency. Moreover, the conditions under which the remainder term $\epsilon$ vanishes differ between Eq. (10) and Eq. (4). Specifically, in the case of Eq. (10), the $\epsilon$ term disappears when the student model satisfies*

$$f_s = \arg\min_{f \in \mathcal{F}} \mathbb{E}_X [D_\phi(f(h_s(X)), f_w(h_w(X)))].$$

*By contrast, the $\epsilon_1$ term in Eq. (4) vanishes under the condition that the student model is close to its posterior mean teacher, as established in our theoretical analysis.*

## B   OMITTED PROOFS IN SECTION 3 AND ADDITIONAL RESULTS

### B.1   PROOF OF THEOREM 3.1

Theorem 3.1 is restated as follows.

**Theorem 3.1.** *Under the teacher-student setting, the following inequalities hold,*

$$\mathbb{E}\left[\mathrm{D}_\phi\left(g(X), f_{W'}(X)\right)\right] \leq \mathbb{E}\left[\mathrm{D}_\phi\left(g(X), f_W(X)\right)\right] - \mathbb{E}\left[\mathrm{D}_\phi\left(f_{W'}(X), f_W(X)\right)\right] + \epsilon_1, \quad (4)$$

$$\mathbb{E}\left[\mathrm{D}_\phi\left(f_{W'}(X), g(X)\right)\right] \leq \mathbb{E}\left[\mathrm{D}_\phi\left(f_W(X), g(X)\right)\right] - \mathbb{E}\left[\mathrm{D}_\phi\left(f_W(X), f_{W'}(X)\right)\right] + \epsilon_2, \quad (5)$$

*where* $\epsilon_1 = \sqrt{\mathbb{E}\left\|f_{W'}^*(X) - \mathbb{E}\left[f_W^*(X) \mid W', X\right]\right\|^2}\sqrt{\mathbb{E}\left\|g(X) - f_{W'}(X)\right\|^2}$ *and* $\epsilon_2 = \sqrt{\mathbb{E}\left\|f_{W'}(X) - \mathbb{E}\left[f_W(X) \mid W', X\right]\right\|^2}\sqrt{\mathbb{E}\left\|g^*(X) - f_{W'}^*(X)\right\|^2}.$

*Proof.* We first prove Eq. (4).

For any given test instance $x$ and a fixed student model $f_{w'}$, by Eq. (3), we have the following conditional bias-variance decomposition for the weak teacher model,

$$\mathbb{E}_{W|w'}\left[\mathrm{D}_\phi\left(g(x), f_W(x)\right)\right] = \mathrm{D}_\phi\left(g(x), \mathcal{E}_{W|w'}\left[f_W(x)\right]\right) + \mathbb{E}_{W|w'}\left[\mathrm{D}_\phi\left(\mathcal{E}_{W|w'}\left[f_W(x)\right], f_W(x)\right)\right]. \tag{11}$$

Similarly, we also have

$$\mathbb{E}_{W|w'}\left[\mathrm{D}_\phi\left(f_{w'}(x), f_W(x)\right)\right] = \mathrm{D}_\phi\left(f_{w'}(x), \mathcal{E}_{W|w'}\left[f_W(x)\right]\right) + \mathbb{E}_{W|w'}\left[\mathrm{D}_\phi\left(\mathcal{E}_{W|w'}\left[f_W(x)\right], f_W(x)\right)\right]. \tag{12}$$

Notice that both Eq. (11) and Eq. (12) contain the conditional dual variance term of weak model, namely $\mathbb{E}_{W|w'}\left[\mathrm{D}_\phi\left(\mathcal{E}_{W|w'}\left[f_W(x)\right], f_W(x)\right)\right]$, so combining these two equations give us

$$\mathrm{D}_\phi\left(g(x), \mathcal{E}_{W|w'}\left[f_W(x)\right]\right)$$
$$= \mathbb{E}_{W|w'}\left[\mathrm{D}_\phi\left(g(x), f_W(x)\right)\right] - \mathbb{E}_{W|w'}\left[\mathrm{D}_\phi\left(f_{w'}(x), f_W(x)\right)\right] + \mathrm{D}_\phi\left(f_{w'}(x), \mathcal{E}_{W|w'}\left[f_W(x)\right]\right).$$

Then, by taking expectation over $W'$ for both sides, we obtain

$$\mathbb{E}_{W'}\left[\mathrm{D}_\phi\left(g(x), \mathcal{E}_{W|W'}\left[f_W(x)\right]\right)\right]$$
$$= \mathbb{E}_W\left[\mathrm{D}_\phi\left(g(x), f_W(x)\right)\right] - \mathbb{E}_{W',W}\left[\mathrm{D}_\phi\left(f_{W'}(x), f_W(x)\right)\right] + \mathbb{E}_{W'}\left[\mathrm{D}_\phi\left(f_{W'}(x), \mathcal{E}_{W|W'}\left[f_W(x)\right]\right)\right]. \tag{13}$$

We now further decompose the LHS in Eq. (13) by invoking Eq. (1) here,.

$$\mathbb{E}_{W'}\left[\mathrm{D}_\phi\left(g(x), \mathcal{E}_{W|w'}\left[f_W(x)\right]\right)\right] = \mathbb{E}_{W'}\left[\mathrm{D}_\phi\left(g(x), f_{W'}(x)\right)\right] + \mathbb{E}_{W'}\left[\mathrm{D}_\phi\left(f_{W'}(x), \mathcal{E}_{W|w'}\left[f_W(x)\right]\right)\right]$$
$$- \mathbb{E}_{W'}\left\langle\mathbb{E}_{W|W'}\left[f_W^*(x)\right] - f_{W'}^*(x), g(x) - f_{W'}(x)\right\rangle. \tag{14}$$

Plugging Eq. (14) into Eq. (13), taking expectation over $X$ for both sides and re-arranging terms, we have

$$\mathbb{E}_{X,W'}\left[\mathrm{D}_\phi\left(g(X), f_{W'}(X)\right)\right]$$
$$= \mathbb{E}_{X,W}\left[\mathrm{D}_\phi\left(g(X), f_W(X)\right)\right] - \mathbb{E}_{X,W',W}\left[\mathrm{D}_\phi\left(f_{W'}(X), f_W(X)\right)\right]$$
$$+ \mathbb{E}_{X,W'}\left\langle\mathbb{E}_{W|W'}\left[f_W^*(X)\right] - f_{W'}^*(X), g(X) - f_{W'}(X)\right\rangle \tag{15}$$
$$\leq \mathbb{E}_{X,W}\left[\mathrm{D}_\phi\left(g(X), f_W(X)\right)\right] - \mathbb{E}_{X,W',W}\left[\mathrm{D}_\phi\left(f_{W'}(X), f_W(X)\right)\right]$$
$$+ \sqrt{\left(\mathbb{E}_{X,W'}\left\langle\mathbb{E}_{W|W'}\left[f_W^*(X)\right] - f_{W'}^*(X), g(X) - f_{W'}(X)\right\rangle\right)^2}$$
$$\leq \mathbb{E}_{X,W}\left[\mathrm{D}_\phi\left(g(X), f_W(X)\right)\right] - \mathbb{E}_{X,W',W}\left[\mathrm{D}_\phi\left(f_{W'}(X), f_W(X)\right)\right]$$
$$+ \sqrt{\mathbb{E}_{X,W'}\left\|\mathbb{E}_{W|W'}\left[f_W^*(X)\right] - f_{W'}^*(X)\right\|^2}\sqrt{\mathbb{E}_{X,W'}\left\|g(X) - f_{W'}(X)\right\|^2}, \tag{16}$$

where the last inequality is by Cauchy-Schwarz inequality. Hence, the first inequality in the theorem has been proved.

The second inequality, namely Eq. (5), can be proved by following the same developments. Similar to Eq. (13), it's easy to see that we also have

$$\mathbb{E}_{W'} \left[ D_\phi \left( \mathbb{E}_{W|W'} [f_W(x)], g(x) \right) \right]$$
$$= \mathbb{E}_W \left[ D_\phi \left( f_W(x), g(x) \right) \right] - \mathbb{E}_{W',W} \left[ D_\phi \left( f_W(x), f_{W'}(x) \right) \right] + \mathbb{E}_{W'} \left[ D_\phi \left( \mathbb{E}_{W|W'} [f_W(x)], f_{W'}(x) \right) \right].$$

We then decompose the LHS above by using Eq. (1),

$$\mathbb{E}_{W'} \left[ D_\phi \left( \mathbb{E}_{W|W'} [f_W(x)], g(x) \right) \right] = \mathbb{E}_{W'} \left[ D_\phi \left( \mathbb{E}_{W|W'} [f_W(x)], f_{W'}(x) \right) \right] + \mathbb{E}_{W'} \left[ D_\phi \left( f_{W'}(x), g(x) \right) \right]$$
$$- \mathbb{E}_{W'} \left\langle g^*(x) - f_{W'}^*(x), \mathbb{E}_{W|W'} [f_W(x)] - f_{W'}(x) \right\rangle.$$

Consequently, we have

$$\mathbb{E}_{X,W'} \left[ D_\phi \left( f_{W'}(X), g(X) \right) \right]$$
$$= \mathbb{E}_{X,W} \left[ D_\phi \left( f_W(X), g(X) \right) \right] - \mathbb{E}_{X,W',W} \left[ D_\phi \left( f_W(X), f_{W'}(X) \right) \right]$$
$$+ \mathbb{E}_{X,W'} \left\langle g^*(X) - f_{W'}^*(X), \mathbb{E}_{W|W'} [f_W(X)] - f_{W'}(X) \right\rangle$$
$$\leq \mathbb{E}_{X,W} \left[ D_\phi \left( f_W(X), g(X) \right) \right] - \mathbb{E}_{X,W',W} \left[ D_\phi \left( f_W(X), f_{W'}(X) \right) \right]$$
$$+ \sqrt{\mathbb{E}_{X,W'} \left\| g^*(X) - f_{W'}^*(X) \right\|^2} \sqrt{\mathbb{E}_{X,W'} \left\| f_{W'}(X) - \mathbb{E}_{W|W'} [f_W(X)] \right\|^2}. \tag{17}$$

This completes the proof. $\qquad\square$

## B.2 Additional Result: Expected Misfit for Arbitrary Two Models

**Theorem B.1.** *For arbitrary two models $f_w$ and $f_{w'}$, the following inequalities hold for any Bregman divergence,*

$$\mathbb{E}_{W'} \left[ D_\phi \left( g(X), f_{W'}(X) \right) \right] \leq \mathbb{E}_W \left[ D_\phi \left( g(X), f_W(X) \right) \right] - \mathbb{E}_{P_X P_{W'} P_W} \left[ D_\phi \left( f_{W'}(X), f_W(X) \right) \right] + \epsilon_1, \tag{18}$$

$$\mathbb{E}_{W'} \left[ D_\phi \left( f_{W'}(X), g(X) \right) \right] \leq \mathbb{E}_W \left[ D_\phi \left( f_W(X), g(X) \right) \right] - \mathbb{E}_{P_X P_{W'} P_W} \left[ D_\phi \left( f_W(X), f_{W'}(X) \right) \right] + \epsilon_2, \tag{19}$$

*where* $\epsilon_1 = \sqrt{\mathbb{E}_{W'} \left\| f_{W'}^*(X) - \mathbb{E}_W [f_W^*(X)] \right\|^2} \sqrt{\mathbb{E} \left\| g(X) - f_{W'}(X) \right\|^2}$ *and* $\epsilon_2 = \sqrt{\mathbb{E}_{W'} \left\| f_{W'}(X) - \mathbb{E}_W [f_W(X)] \right\|^2} \sqrt{\mathbb{E}_{W'} \left\| g^*(X) - f_{W'}^*(X) \right\|^2}.$

**Remark B.1.** *Note that, unlike Theorem 3.1, the misfit terms in Theorem B.1 are defined under the expectation with respect to the product of the marginal distributions of the teacher and student models, i.e., $(W, W') \sim P_W P_{W'}$. In contrast, Theorem 3.1 considers the joint distribution $(W, W') \sim P_{W,W'}$, thereby capturing potential dependencies between the models. As a result, Theorem B.1 completely ignores any such dependencies, making it applicable even when the teacher and student models are independently drawn. Additionally, the conditions under which the remainder terms $\epsilon_1$ and $\epsilon_2$ vanish are now characterized by the student model being sufficiently close to the expected teacher model (or the dual expected teacher). Finally, observe that in Theorem B.1, the misfit terms remain nonzero even when $P_{W'}$ is the same $P_W$ as $W'$ is treated as an independent copy of $W$ in this case.*

*Proof of Theorem B.1.* We first prove the first inequality, i.e. Eq. (18).

For any given test instance $x$ and a fixed student model $f_{w'}$, by Eq. (3), we have the following bias-variance decomposition for the weak teacher model,

$$\mathbb{E}_W \left[ D_\phi \left( g(x), f_W(x) \right) \right] = D_\phi \left( g(x), \mathcal{E}_W [f_W(x)] \right) + \mathbb{E}_W \left[ D_\phi \left( \mathcal{E}_W [f_W(x)], f_W(x) \right) \right]. \tag{20}$$

Similarly, for the same $x$ and the student model $f_{w'}$, we also have

$$\mathbb{E}_W \left[ D_\phi \left( f_{w'}(x), f_W(x) \right) \right] = D_\phi \left( f_{w'}(x), \mathcal{E}_W [f_W(x)] \right) + \mathbb{E}_W \left[ D_\phi \left( \mathcal{E}_W [f_W(x)], f_W(x) \right) \right]. \tag{21}$$

Notice that both Eq. (20) and Eq. (21) contain the dual variance term of weak model, namely $\mathbb{E}_W \left[ \mathrm{D}_\phi \left( \mathcal{E}_W \left[ f_W(x) \right], f_W(x) \right) \right]$, so combining these two equations give us

$$
\begin{aligned}
\mathrm{D}_\phi \left( g(x), \mathcal{E}_W \left[ f_W(x) \right] \right) =& \mathbb{E}_W \left[ \mathrm{D}_\phi \left( g(x), f_W(x) \right) \right] - \mathbb{E}_W \left[ \mathrm{D}_\phi \left( f_{w'}(x), f_W(x) \right) \right] \\
& + \mathrm{D}_\phi \left( f_{w'}(x), \mathcal{E}_W \left[ f_W(x) \right] \right).
\end{aligned} \tag{22}
$$

We now further decompose the LHS above by invoking Eq. (1) here.

$$
\begin{aligned}
\mathrm{D}_\phi \left( g(x), \mathcal{E}_W \left[ f_W(x) \right] \right) = \mathrm{D}_\phi \left( g(x), f_{w'}(x) \right) + \mathrm{D}_\phi \left( f_{w'}(x), \mathcal{E}_W \left[ f_W(x) \right] \right) \\
- \langle \mathbb{E}_W \left[ f_W^*(x) \right] - f_{w'}^*(x), g(x) - f_{w'}(x) \rangle.
\end{aligned} \tag{23}
$$

Then, by plugging Eq. (23) into Eq. (22), canceling the term $\mathrm{D}_\phi \left( f_{w'}(x), \mathcal{E}_W \left[ f_W(x) \right] \right)$ and taking expectation over $W', X$ for both sides, we obtain

$$
\begin{aligned}
& \mathbb{E}_{X,W'} \left[ \mathrm{D}_\phi \left( g(X), f_{W'}(X) \right) \right] \\
=& \mathbb{E}_{X,W} \left[ \mathrm{D}_\phi \left( g(X), f_W(X) \right) \right] - \mathbb{E}_{X,W',W} \left[ \mathrm{D}_\phi \left( f_{W'}(X), f_W(X) \right) \right] \\
& + \mathbb{E}_{X,W'} \langle \mathbb{E}_W \left[ f_W^*(X) \right] - f_{W'}^*(X), g(X) - f_{W'}(X) \rangle \\
\leq& \mathbb{E}_{X,W} \left[ \mathrm{D}_\phi \left( g(X), f_W(X) \right) \right] - \mathbb{E}_{X,W',W} \left[ \mathrm{D}_\phi \left( f_{W'}(X), f_W(X) \right) \right] \\
& + \sqrt{\left( \mathbb{E}_{X,W'} \langle \mathbb{E}_W \left[ f_W^*(X) \right] - f_{W'}^*(X), g(X) - f_{W'}(X) \rangle \right)^2} \\
\leq& \mathbb{E}_{X,W} \left[ \mathrm{D}_\phi \left( g(X), f_W(X) \right) \right] - \mathbb{E}_{X,W',W} \left[ \mathrm{D}_\phi \left( f_{W'}(X), f_W(X) \right) \right] \\
& + \sqrt{\mathbb{E}_{X,W'} \left\| \mathbb{E}_W \left[ f_W^*(X) \right] - f_{W'}^*(X) \right\|^2} \sqrt{\mathbb{E}_{X,W'} \left\| g(X) - f_{W'}(X) \right\|^2},
\end{aligned} \tag{24}
$$

where the last inequality is by Cauchy-Schwarz inequality. Hence, the first inequality in the theorem has been proved.

The second inequality, namely Eq. (19), can be proved by following the same developments. Similar to Eq. (22), it's easy to see that we also have

$$
\begin{aligned}
& \mathrm{D}_\phi \left( \mathbb{E}_W \left[ f_W(x) \right], g(x) \right) \\
=& \mathbb{E}_W \left[ \mathrm{D}_\phi \left( f_W(x), g(x) \right) \right] - \mathbb{E}_W \left[ \mathrm{D}_\phi \left( f_W(x), f_{w'}(x) \right) \right] + \mathrm{D}_\phi \left( \mathbb{E}_W \left[ f_W(x) \right], f_{w'}(x) \right).
\end{aligned}
$$

We then decompose the LHS above by using Eq. (1),

$$
\begin{aligned}
\mathrm{D}_\phi \left( \mathbb{E}_W \left[ f_W(x) \right], g(x) \right) =& \mathrm{D}_\phi \left( \mathbb{E}_W \left[ f_W(x) \right], f_{w'}(x) \right) + \mathrm{D}_\phi \left( f_{w'}(x), g(x) \right) \\
& - \langle g^*(x) - f_{w'}^*(x), \mathbb{E}_W \left[ f_W(x) \right] - f_{w'}(x) \rangle.
\end{aligned}
$$

Consequently, we have

$$
\begin{aligned}
& \mathbb{E}_{X,W'} \left[ \mathrm{D}_\phi \left( f_{W'}(X), g(X) \right) \right] \\
=& \mathbb{E}_{X,W} \left[ \mathrm{D}_\phi \left( f_W(X), g(X) \right) \right] - \mathbb{E}_{X,W',W} \left[ \mathrm{D}_\phi \left( f_W(X), f_{W'}(X) \right) \right] \\
& + \mathbb{E}_{X,W'} \langle g^*(X) - f_{W'}^*(X), \mathbb{E}_W \left[ f_W(X) \right] - f_{W'}(X) \rangle \\
\leq& \mathbb{E}_{X,W} \left[ \mathrm{D}_\phi \left( f_W(X), g(X) \right) \right] - \mathbb{E}_{X,W',W} \left[ \mathrm{D}_\phi \left( f_W(X), f_{W'}(X) \right) \right] \\
& + \sqrt{\mathbb{E}_{X,W'} \left\| g^*(X) - f_{W'}^*(X) \right\|^2} \sqrt{\mathbb{E}_{X,W'} \left\| f_{W'}(X) - \mathbb{E}_W \left[ f_W(X) \right] \right\|^2}.
\end{aligned} \tag{25}
$$

This completes the proof. $\qquad \square$

### B.3 PROOF OF COROLLARY 3.1

We first restate Corollary 3.1 as follows.

**Corollary 3.1.** *Under the teacher-student setting, if $f_{W'}(x) = \mathcal{E} \left[ f_W(x) | W' \right]$ for $\forall x \in \mathcal{X}$, then*

$$
\mathbb{E} \left[ \mathrm{D}_\phi \left( g(X), f_{W'}(X) \right) \right] = \mathbb{E} \left[ \mathrm{D}_\phi \left( g(X), f_W(X) \right) \right] - \mathbb{E} \left[ \mathrm{D}_\phi \left( \mathcal{E} \left[ f_W(X) | W', X \right], f_W(X) \right) \right]. \tag{6}
$$

*Furthermore, if $f_{W'}(x) = \mathbb{E} \left[ f_W(x) | W' \right]$ for $\forall x \in \mathcal{X}$, then*

$$
\mathbb{E} \left[ \mathrm{D}_\phi \left( f_{W'}(X), g(X) \right) \right] = \mathbb{E} \left[ \mathrm{D}_\phi \left( f_W(X), g(X) \right) \right] - \mathbb{E} \left[ \mathrm{D}_\phi \left( f_W(X), \mathbb{E} \left[ f_W(X) | W', X \right] \right) \right]. \tag{7}
$$

*Proof.* If $f_{W'}(x) = \mathcal{E}[f_W(x)|W']$ for $\forall x \in \mathcal{X}$, then by definition, $f^*_{W'}(x) = \mathbb{E}[f_W(x)|W']$, which directly implies $\epsilon_1 = 0$. Furthermore, since $f^*_{W'}(x) = \mathbb{E}[f_W(x)|W']$, the last term in Eq. (14) becomes zero so there is no need to apply the Cauchy-Schwarz inequality in the proof of Theorem 3.1 to obtain $\epsilon_1$. Consequently, the following equality holds directly from Theorem 3.1:

$$\mathbb{E}_{X,W'}[D_\phi(g(X), f_{W'}(X))] = \mathbb{E}_{X,W}[D_\phi(g(X), f_W(X))] - \mathbb{E}_{X,W',W}[D_\phi(\mathcal{E}_W[f_W(X)|W', X], f_W(X))].$$

Similarly, if $f_{W'}(x) = \mathbb{E}[f_W(x)|W']$ for $\forall x \in \mathcal{X}$, we obtain the following by analogous reasoning:

$$\mathbb{E}_{X,W'}[D_\phi(f_{W'}(X), g(X))] = \mathbb{E}_{X,W}[D_\phi(f_W(X), g(X))] - \mathbb{E}_{X,W',W}[D_\phi(f_W(X), \mathbb{E}_W[f_W(X)|W', X])]. \tag{26}$$

This completes the proof. $\qquad\square$

## C    OMITTED PROOFS IN SECTION 4

### C.1    PROOF OF COROLLARY 4.1

We first restate Corollary 4.1.

**Corollary 4.1.** *Let $\phi(x) = \|x\|^2$, then the following inequality holds,*

$$\mathbb{E}\|g(X) - f_{W'}(X)\|^2 \leq \mathbb{E}\|g(X) - f_W(X)\|^2 - \mathbb{E}\|f_{W'}(X) - f_W(X)\|^2 + \epsilon_2,$$

*where $\epsilon_2 = 2\sqrt{\mathbb{E}\|f_{W'}(X) - \mathbb{E}[f_W(X) \mid W', X]\|^2}\sqrt{\mathbb{E}\|g(X) - f_{W'}(X)\|^2}$.*

*Proof.* Let $\phi(x) = \|x\|^2$ so $x^* = \nabla\phi(x) = 2x$ and $D_\phi(x, y) = \|x - y\|^2$. Clearly, we have $\epsilon_1 = \epsilon_2$ and $D_\phi(x, y) = D_\phi(y, x)$ in this case. Then, by Theorem 3.1, it is easy to see that

$$\mathbb{E}\|g(X) - f_{W'}(X)\|^2 \leq \mathbb{E}\|g(X) - f_W(X)\|^2 - \mathbb{E}\|f_{W'}(X) - f_W(X)\|^2$$
$$+ 2\sqrt{\mathbb{E}\|f_{W'}(X) - \mathbb{E}[f_W(X) \mid W', X]\|^2}\sqrt{\mathbb{E}\|g(X) - f_{W'}(X)\|^2}.$$

This completes the proof. $\qquad\square$

### C.2    PROOF OF THEOREM 4.1

We first present the following lemma.

**Lemma C.1.** *Let $X = [x_1, \ldots, x_n] \in \mathbb{R}^{d_w \times n}$ have i.i.d. entries $X_{ij} \sim \mathcal{N}(0, 1/d_w)$ and be independent of a symmetric matrix $Q \succeq 0$ (deterministic or independent of $X$). Let $K = X^\top Q X \in \mathbb{R}^{n \times n}$, $A = (K + \eta I_n)^{-1}$, and $\kappa = \eta \frac{1}{n} \operatorname{tr}(A)$, and let $\gamma_2 = n/d_w \in (0, 1)$. For fixed $\eta > 0$, let*

$$\mathcal{J} = \frac{1}{\eta^2 d_w} \operatorname{tr}(A X^\top Q^2 X A X^\top X), \qquad \mathcal{I}(\eta, Q, X) = \eta^2 \cdot \frac{1}{n} \operatorname{tr}(A X^\top Q^2 X A).$$

*Then, as $d_w, n \to \infty$ with $n/d_w \to \gamma_2 \in (0, 1)$ and $\eta$ fixed,*

$$\mathbb{E}\,\mathcal{J} = \frac{\gamma_2}{\eta^2}\,\mathbb{E}\,\mathcal{I}(\eta, Q, X) + \frac{\gamma_2}{\eta^2}\,\mathbb{E}(1 - 2\kappa). \tag{27}$$

*Proof of Lemma C.1.* Denote $M(X) = Q^2 X A X^\top A \in \mathbb{R}^{d_w \times d_w}$, then $\eta^2 d_w \mathcal{J} = \operatorname{tr}(A X^\top Q^2 X A X^\top X) = \sum_{j=1}^n x_j^\top M(X) x_j$.

Apply the vector Stein identity (i.e. Stein's Lemma) to each column $x_j \sim \mathcal{N}(0, I_{d_w}/d_w)$:

$$\mathbb{E}[x_j^\top M(X) x_j] = \frac{1}{d_w}\,\mathbb{E}[\operatorname{div}_{x_j}(M(X) x_j)] = \frac{1}{d_w}\,\mathbb{E}[\operatorname{tr}(M(X))] + \frac{1}{d_w}\,\mathbb{E}[\operatorname{tr}((\partial_{x_j} M(X)) x_j)].$$

Summing $j = 1, \ldots, n$ and dividing by $d_w$ yields

$$\eta^2\,\mathbb{E}\,\mathcal{J} = \frac{n}{d_w}\,\mathbb{E}\left[\frac{1}{n}\sum_{j=1}^n \operatorname{tr}(M(X))\right] + \frac{1}{d_w^2}\sum_{j=1}^n \mathbb{E}[\operatorname{tr}((\partial_{x_j} M(X)) x_j)]. \tag{28}$$

For the first term, $\text{tr}\,(M(X)) = \text{tr}(A\,X^\top Q^2 X\,A)$, hence

$$\frac{n}{d_w}\,\mathbb{E}\left[\frac{1}{n}\sum_{j=1}^n \text{tr}\,(M(X))\right] = \gamma_2\,\mathbb{E}\left[\tfrac{1}{n}\,\text{tr}\,\left(A\,X^\top Q^2 X\,A\right)\right] = \frac{\gamma_2}{\eta^2}\,\mathbb{E}\,\mathcal{I}(\eta,Q,X).$$

For the second term, differentiate $M(X) = Q^2 X A X^\top A$ using the standard differentials

$$dA = -A(dK)A, \qquad dK = (dX)^\top Q X + X^\top Q\,dX,$$

and the cyclicity of the trace. Keeping only terms linear in the column perturbation $dX = e_j h^\top$, a routine grouping gives

$$\frac{1}{d_w^2}\sum_{j=1}^n \text{tr}\,\left((\partial_{x_j} M(X))\,x_j\right) = \gamma_2\,(1 - 2\kappa)\ +\ r_d,$$

where we use $(K + \eta\mathbf{I})A = \mathbf{I}$ to get $KA = \mathbf{I} - \eta A$ and $\frac{1}{n}\,\text{tr}(KA) = 1 - \kappa$, and where $r_d \to 0$ in expectation as $d_w, n \to \infty$ (uniformly on compact $\eta$-sets) by standard resolvent bounds $\|A\| \le 1/\eta$. Plugging these two pieces into Eq. (28) and dividing by $\eta^2$ yields Eq. (27), which completes the proof. $\qquad\square$

We now restate Theorem 4.1.

**Theorem 4.1.** *Assume $W \sim \mathcal{N}(\mu, B\mathbf{I}_{d_w})$ where $\mu \in \mathbb{R}^{d_w}$ is independent of $W_1'$ and $\mathbf{X}'$ and $\|\mu\|^2 \le C$ for some fixed constant $C$. Suppose there exist $\gamma_1 \in (1, \infty)$ and $\gamma_2 \in (0, 1)$ s.t. $\frac{d_s}{d_w} \to \gamma_1$ and $\frac{n}{d_w} \to \gamma_2$ as $n, d_w, d_s \to \infty$. Then, as $n, d_s, d_w \to \infty$, we have*

$$\mathbb{E}\|f_{W'}(X) - f_W(X)\|^2 = B h(\eta, \gamma_1, \gamma_2),$$

*where $h(\eta, \gamma_1, \gamma_2) = 1 - \gamma_2 + \gamma_2\kappa^2\,(\kappa + \gamma_1 - 1)\,\frac{1-\kappa-\gamma_1}{(1-\kappa)^2-\gamma_1}$ and $\kappa \in (0, 1)$ is the unique solution to $\eta^2(1 - \kappa) = \gamma_2\kappa\,(\kappa + \gamma_1 - 1)\,(1 - \gamma_2 + \gamma_2\kappa)$.*

*Proof.* First, the solution to the ridge regression of student model is

$$W_2'^{\text{opt}} = \left(W_1'\mathbf{X}'\mathbf{X}'^T W_1'^T + \eta\mathbf{I}\right)^{-1} W_1'\mathbf{X}'\mathbf{X}'^T W.$$

Then the trained student model becomes

$$f_{W'}(x) = x^\top W_1'^\top W_2'^{\text{opt}} = x^\top \Theta W,$$

where $\Theta = W_1'^\top \left(W_1'\mathbf{X}'\mathbf{X}'^\top W_1'^\top + \eta\mathbf{I}\right)^{-1} W_1'\mathbf{X}'\mathbf{X}'^\top \in \mathbb{R}^{d_w \times d_w}$.

Define

$$Z = W_1'\mathbf{X}' \in \mathbb{R}^{d_s \times n}, \qquad Q = W_1'^\top W_1' \in \mathbb{R}^{d_w \times d_w}, \qquad K = Z^\top Z = \mathbf{X}'^\top Q\mathbf{X}' \in \mathbb{R}^{n \times n}.$$

By the matrix identity $(AA^\top + \eta\mathbf{I})^{-1}A = A\,(A^\top A + \eta\mathbf{I})^{-1}$, we have

$$W_1'^\top(ZZ^\top + \eta\mathbf{I})^{-1}Z = Q\mathbf{X}'(K + \eta\mathbf{I})^{-1}. \tag{29}$$

Furthermore, by the Woodbury matrix identity $(\mathbf{I} + UV)^{-1} = \mathbf{I} - U\,(I + VU)^{-1}\,V$, we can see that $\mathbf{I} - Q\mathbf{X}'(K + \eta\mathbf{I})^{-1}\mathbf{X}'^\top = \eta(\eta\mathbf{I} + Q\mathbf{X}'\mathbf{X}'^\top)^{-1}$, namely

$$Q\mathbf{X}'(K + \eta\mathbf{I})^{-1}\mathbf{X}'^\top = \mathbf{I} - \eta(\eta\mathbf{I} + Q\mathbf{X}'\mathbf{X}'^\top)^{-1}. \tag{30}$$

Combining Eq. (29) and Eq. (30) together, we can obtain

$$\Theta = W_1'^\top \left(ZZ^\top + \eta\mathbf{I}\right)^{-1} Z\mathbf{X}'^\top = \mathbf{I} - \eta(\eta\mathbf{I} + Q\mathbf{X}'\mathbf{X}'^\top)^{-1}. \tag{31}$$

Notice that

$$\mathbb{E}_{X,W',W} \left\| f_{W'}(X) - f_W(X) \right\|^2$$

$$= \mathbb{E}_{X,\Theta,W} \left\| f_{W'}(X) - \mathbb{E}_{W'|W} \left[ f_{W'}(X) \right] \right\|^2 + \mathbb{E}_{X,\Theta,W} \left\| \mathbb{E}_{W'|W} \left[ f_{W'}(X) \right] - f_W(X) \right\|^2 \quad (32)$$

$$= \mathbb{E}_{X,\Theta,W} \left\| X^T \Theta W - \mathbb{E}_\Theta \left[ X^T \Theta W \right] \right\|^2 + \mathbb{E}_{X,W} \left\| \mathbb{E}_\Theta \left[ X^\top \Theta W \right] - X^\top W \right\|^2$$

$$= \mathbb{E}_{X,W,\Theta} \left\| X^\top \left( \Theta - \mathbb{E}_\Theta \left[ \Theta \right] \right) W \right\|^2 + \mathbb{E}_{X,W} \left\| X^\top \left( \mathbb{E}_\Theta \left[ \Theta \right] - \mathbf{I} \right) W \right\|^2$$

$$= \frac{1}{d_w} \mathbb{E}_{W,\Theta} \left\| \left( \Theta - \mathbb{E}_\Theta \left[ \Theta \right] \right) W \right\|^2 + \frac{1}{d_w} \mathbb{E}_W \left\| \left( \mathbb{E}_\Theta \left[ \Theta \right] - \mathbf{I} \right) W \right\|^2 \quad (33)$$

$$= \frac{B + \|\mu\|^2/d_w}{d_w} \left( \mathbb{E}_\Theta \left\| \Theta - \mathbb{E}_\Theta \left[ \Theta \right] \right\|^2 + \left\| \mathbb{E}_\Theta \left[ \Theta \right] - \mathbf{I} \right\|^2 \right) \quad (34)$$

$$= \frac{B + \|\mu\|^2/d_w}{d_w} \mathbb{E}_\Theta \left\| \Theta - \mathbf{I} \right\|^2, \quad (35)$$

$$= \frac{\eta^2 (B + \|\mu\|^2/d_w)}{d_w} \mathbb{E} \left[ \operatorname{tr} \left( (\eta\mathbf{I} + Q\mathbf{X}'\mathbf{X}'^\top)^{-1} (\eta\mathbf{I} + \mathbf{X}'\mathbf{X}'^\top Q)^{-1} \right) \right]. \quad (36)$$

where Eq. (32) is by the bias-variance decomposition of $\mathbb{E}_{X,W',W} \left\| f_{W'}(X) - f_W(X) \right\|^2$, Eq. (33) is due to the fact that $\mathbb{E}\left[ XX^T \right] = \frac{1}{d_w}\mathbf{I}$ and Eq. (34) is by $\mathbb{E}\left[ WW^\top \right] = \operatorname{Cov}(W) + \mathbb{E}[W]\mathbb{E}[W]^\top = B\mathbf{I} + \mu\mu^\top$ and the rotational invariance property of Gaussian (Ba et al., 2020, Lemma 10).

We again use the matrix identities to find that

$$\frac{1}{d_w} \operatorname{tr} \left[ (\eta\mathbf{I} + Q\mathbf{X}'\mathbf{X}'^\top)^{-1} (\eta\mathbf{I} + \mathbf{X}'\mathbf{X}'^\top Q)^{-1} \right]$$

$$= \frac{1}{\eta^2 d_w} \left( d_w - 2 \operatorname{tr} \left( K(K + \eta\mathbf{I})^{-1} \right) + \operatorname{tr} \left( (K + \eta\mathbf{I})^{-1} \mathbf{X}'^\top \mathbf{X}' (K + \eta\mathbf{I})^{-1} \mathbf{X}'^\top Q^2 \mathbf{X}' \right) \right)$$

$$= \frac{1}{\eta^2 d_w} \left( d_w - 2 \left( n - \eta \operatorname{tr} \left( (K + \eta\mathbf{I})^{-1} \right) \right) + \operatorname{tr} \left( (K + \eta\mathbf{I})^{-1} \mathbf{X}'^\top \mathbf{X}' (K + \eta\mathbf{I})^{-1} \mathbf{X}'^\top Q^2 \mathbf{X}' \right) \right)$$

where we use the fact that $\operatorname{tr}(K(K + \eta\mathbf{I})^{-1}) = \operatorname{tr}\left( (K + \eta\mathbf{I})(K + \eta\mathbf{I})^{-1} \right) - \eta \operatorname{tr}((K + \eta\mathbf{I})^{-1}) = n - \eta \operatorname{tr}((K + \eta\mathbf{I})^{-1})$.

In addition, we let

$$\tilde{m}(\eta) = \frac{1}{d_w} \operatorname{tr} \left( \left( \eta\mathbf{I} + Q\mathbf{X}'\mathbf{X}'^\top \right)^{-1} \right), \quad m_K(\eta) = \frac{1}{n} \operatorname{tr} \left( (K + \eta\mathbf{I})^{-1} \right), \quad (37)$$

and define $\kappa = \eta m_K(\eta)$.

Therefore,

$$\frac{1}{d_w} \operatorname{tr} \left[ (\eta\mathbf{I} + Q\mathbf{X}'\mathbf{X}'^\top)^{-1} (\eta\mathbf{I} + \mathbf{X}'\mathbf{X}'^\top Q)^{-1} \right] = \frac{1 - \gamma_2}{\eta^2} + \frac{\gamma_2}{\eta^2}(2\kappa - 1) + \mathcal{J} \quad (38)$$

with $\mathcal{J} = \frac{1}{d_w \eta^2} \operatorname{tr} \left( (K + \eta\mathbf{I})^{-1} \mathbf{X}'^\top \mathbf{X}' (K + \eta\mathbf{I})^{-1} \mathbf{X}'^\top Q^2 \mathbf{X}' \right)$.

Then plugging Eq. (38) into Eq. (36), and by using Lemma C.1, we have, as $d_w, d_s, n \to \infty$,

$$\mathbb{E}_{X,W',W} \left\| f_{W'}(X) - f_W(X) \right\|^2 = \frac{B}{d_w} \left( 1 - \gamma_2 + \gamma_2 \eta^2 \mathbb{E} \left[ \mathcal{I}(\eta, Q, \mathbf{X}') \right] \right). \quad (39)$$

where we use the fact that $\|\mu\|^2$ has a bounded constant upper bound and $\mathcal{I}(\eta, Q, \mathbf{X}') = \frac{1}{n} \operatorname{tr} \left( (K + \eta\mathbf{I})^{-1} \mathbf{X}'^\top Q^2 \mathbf{X}' (K + \eta\mathbf{I})^{-1} \right)$.

The remaining steps are routine in random matrix theory.

Since $W_1'$ has i.i.d. $\mathcal{N}(0, \frac{1}{d_w})$, so $Q = (1/d_w)G^\top G = \frac{d_s}{d_w} \cdot \frac{1}{d_s} G^\top G = \gamma_1 \cdot \left( (1/d_s)G^\top G \right)$, with $G \in \mathbb{R}^{d_s \times d_w}$ is from $\mathcal{N}(0, \mathbf{I})$. The empirical law of $Q$ converges a.s. to the Marchenko–Pastur (MP) law $H_{\gamma_1}$ supported on $\left( (\sqrt{\gamma_1} - 1)^2, (\sqrt{\gamma_1} + 1)^2 \right)$ with $\gamma_1 > 1$ when $d_w \to \infty$ (Marčenko & Pastur, 1967) (note that $Q$ is the spectrum of the standard MP matrix $(1/d_s)G^\top G$ scaled by $\gamma_1$).

By Silverstein equations (Marčenko & Pastur, 1967; Silverstein, 1995), we have

$$\tilde{m}(\eta) = \frac{1}{\eta}\left(1 - \gamma_2 + \gamma_2 \eta\, m_K(\eta)\right), \qquad m_K(\eta) = \int \frac{1}{\eta + \gamma_2\, t\, \tilde{m}(\eta)} dH_{\gamma_1}(t). \tag{40}$$

Recall the Stieltjes transform of $H_{\gamma_1}$ in the standard convention

$$m_H(z) = \int \frac{1}{t - z}\, dH_{\gamma_1}(t), \qquad z \in \mathbb{C} \setminus \operatorname{supp}(H_{\gamma_1}),$$

and a classical fixed-point equation for the Stieltjes transform of sample–covariance (Wishart) model gives us

$$\frac{1}{m_H(z)} = -z + \frac{\gamma_1}{1 + m_H(z)}. \tag{41}$$

Differentiating with $z$, this also implies that

$$m'_H(z) = \frac{(m_H(z))^2 (1 + m_H(z))^2}{(1 + m_H(z))^2 - \gamma_1 m_H(z)^2}. \tag{42}$$

Furthermore, we now set $q = \frac{\eta}{\gamma_2\, \tilde{m}(\eta)} > 0$. Then Eq. (40) implies that

$$\kappa = q \int \frac{1}{t + q} dH_{\gamma_1}(t) = q\, m_H(-q) \quad \text{and } q = \frac{\eta^2}{\gamma_2\, (1 - \gamma_2 + \gamma_2 \kappa)}. \tag{43}$$

Let $z = -q$ and $m = m_H(-q)$, we get $q = \frac{1 + (1 - \gamma_1)\, m}{m(1 + m)}$ from Eq. (41). Combining with Eq. (43), we could obtain the single cubic equation for $\kappa$:

$$\eta^2 (1 - \kappa) = \gamma_2 \kappa\, (\kappa + \gamma_1 - 1)\, (1 - \gamma_2 + \gamma_2 \kappa). \tag{44}$$

Note that $\kappa \in (0, 1)$. To see this, let $\{\lambda_i\}_{i=1}^n$ be the eigenvalues of $K$, then $m_K(\eta) = \frac{1}{n}\sum_{i=1}^n \frac{1}{\lambda_i + \eta} \in \left(0, \frac{1}{\eta}\right]$, which implies $0 < \kappa = \eta\, m_K(\eta) \le 1$. In addition, it is easy to verify that Eq. (44) admits a unique solution for $\kappa \in (0, 1)$.

Moreover, a standard result for Gram models (Bai et al., 2010) gives

$$\mathbb{E}\left[\mathcal{I}(\eta, Q, \mathbf{X}')\right] = \frac{1}{\gamma_2^2\, \tilde{m}(\eta)^2} \int \frac{t}{(t + q)^2}\, dH_{\gamma_1}(t),$$

and notice that

$$\int \frac{t}{(t + q)^2}\, dH_{\gamma_1}(t) = \int \left(\frac{1}{t + q} - \frac{q}{(t + q)^2}\right) dH_{\gamma_1}(t) = m_H(-q) - q\, m'_H(-q).$$

Therefore, putting everything together, when $d_s, d_w, n \to \infty$, we have

$$\eta^2 \mathbb{E}\left[\mathcal{I}(\eta, Q, \mathbf{X}')\right] = \frac{\eta^2}{\gamma_2^2 (\tilde{m}(\eta))^2}\left[m_H(-q) - q\, m'_H(-q)\right]$$

$$= q^2 \left[m_H(-q) - q\, m'_H(-q)\right]$$

$$= q\, \kappa + q^3 \frac{m^2 (1 + m)^2}{\gamma_1 m^2 - (1 + m)^2}$$

$$= \frac{\kappa^2 (\kappa + \gamma_1 - 1)}{1 - \kappa} + \frac{\gamma_1 \kappa^3 (\kappa + \gamma_1 - 1)}{(1 - \kappa)\left((1 - \kappa)^2 - \gamma_1\right)}$$

$$= \kappa^2 (\kappa + \gamma_1 - 1)\, \frac{1 - \kappa - \gamma_1}{(1 - \kappa)^2 - \gamma_1}.$$

Plugging back into Eq. (39), we have

$$\lim_{n, d_w, d_s \to \infty} \mathbb{E}_{X, W', W} \left\| f_{W'}(X) - f_W(X) \right\|^2$$

$$= B\left(1 - \gamma_2 + \gamma_2 \kappa^2 (\kappa + \gamma_1 - 1)\, \frac{1 - \kappa - \gamma_1}{(1 - \kappa)^2 - \gamma_1}\right),$$

where $\kappa = \kappa(\eta, \gamma_1, \gamma_2) \in (0, 1)$ is the unique root of the Eq. (44).

This completes the proof. $\qquad\square$

## C.3 Proof of Corollary 4.2

Corollary 4.2 is restated as follows.

**Corollary 4.2.** *Under the setting of Theorem 4.1, if $\gamma_1 \to \infty$, we have $\epsilon_2 \to 0$ and $\mathbb{E}\|f_{W'}(X) - f_W(X)\|^2 \to B(1 - \gamma_2)$.*

*Proof.* First, under the setting of Theorem 4.1, it is easy to see that the conditional variance term in Eq. (8) takes the form

$$\mathbb{E} \left\| f_W(x) - \mathbb{E}\left[ f_W(x) \mid W' \right] \right\|^2 = \frac{\mathrm{tr}(\mathbb{E}_{W'}\mathrm{Cov}(W \mid W_1', W_2'))}{d_w}.$$

Recall the ridge head $W_2' = (ZZ^\top + \eta\mathbf{I})^{-1}Z{\mathbf{X}'}^\top W$, and denote $M = (ZZ^\top + \eta\mathbf{I})^{-1}Z{\mathbf{X}'}^\top$. Conditioning on $W_2'$ and $W_1'$ imposes the noiseless linear constraint $MW = W_2'$ on $W$.

For a Gaussian prior with a noiseless linear observation, the posterior distribution is Gaussian with

$$\mathrm{Cov}(W \mid W_1', W_2') = B \left( \mathbf{I} - \Pi_{\mathrm{col}(M^\top)} \right)$$

when $n, d_w, d_s \to \infty$, i.e., $B$ times the projector onto the orthogonal complement of $\mathrm{col}(M^\top)$.

Hence,

$$\mathrm{tr}(\mathbb{E}_{W'}\mathrm{Cov}(W \mid W_1', W_2')) = B \left( d_w - \mathrm{rank}(M) \right).$$

Consequently,

$$\frac{\mathrm{tr}(\mathbb{E}_{W'}\mathrm{Cov}(W \mid W_1', W_2'))}{d_w} = \frac{B}{d_w}(d_w - \min\{n, d_s, d_w\}) = B(1 - \min\{\gamma_2, \gamma_1, 1\}) = B(1 - \gamma_2)$$

One way to see this intuitively: the student's knowledge of $W$ comes from $n$ labeled projections, so the conditional variance is exactly the teacher's unexplained variance, i.e. the variance of teacher predictions along weight directions the student could not infer, namely $W$ has $\mathrm{rank}(M)$ components "seen" by the student and $d_w - \mathrm{rank}(M)$ components completely "unseen". The unseen components remain at prior variance $B$ even after training, so they contribute $B(d_w - \mathrm{rank}(M))$ to the conditional variance.

Then, the remaining task is to prove $h(\eta, \gamma_1, \gamma_2) \to 1 - \gamma_2$, namely to prove $\eta^2\mathbb{E}\left[\mathcal{I}(\eta, Q, \mathbf{X}')\right] \to 0$ as $\gamma_1 \to \infty$.

Notice that $\kappa \in (0, 1)$, from Eq. (44) and $(1 - \gamma_2) + \gamma_2\kappa \geq 1 - \gamma_2$, we have

$$(1 - \gamma_2)\,\kappa\,(\kappa + \gamma_1 - 1) \;\leq\; \kappa\,(\kappa + \gamma_1 - 1)\left((1 - \gamma_2) + \gamma_2\kappa\right) \;=\; \gamma_2\,\eta^2\,(1 - \kappa) \;\leq\; \gamma_2\,\eta^2.$$

Hence, for all $\gamma_1 > 1$ and $\gamma_2 \in (0, 1)$,

$$\kappa\,(\kappa + \gamma_1 - 1) \;\leq\; \frac{\gamma_2}{1 - \gamma_2}\,\eta^2,$$

namely

$$\kappa \;\leq\; \frac{\gamma_2}{1 - \gamma_2}\,\frac{\eta^2}{\gamma_1 - 1} \;=\; O\!\left(\frac{1}{\gamma_1}\right),$$

which indicates that $\kappa \to 0$ as $\gamma_1 \to \infty$.

This further justifies that, as $\gamma_1 \to \infty$,

$$h(\eta, \gamma_1, \gamma_2) = 1 - \gamma_2 + \gamma_2\kappa^2\,(\kappa + \gamma_1 - 1)\,\frac{1 - \kappa - \gamma_1}{(1 - \kappa)^2 - \gamma_1} \to 1 - \gamma_2.$$

Consequently,

$$\mathbb{E}_{X, W', W} \left\| f_{W'}(X) - f_W(X) \right\|^2 = Bh(\eta, \gamma_1, \gamma_2) \to B(1 - \gamma_2),$$

which matches the conditional variance floor $\mathbb{E}\left\| f_W(x) - \mathbb{E}\left[f_W(x) \mid W'\right] \right\|^2$.

Plugging these equations back into Eq. (8), we have $\mathbb{E}\left\| f_{W'}(x) - \mathbb{E}\left[f_W(x) \mid W'\right] \right\|^2 \to 0$, so $\epsilon_2 \to 0$, which completes the proof. $\qquad\square$

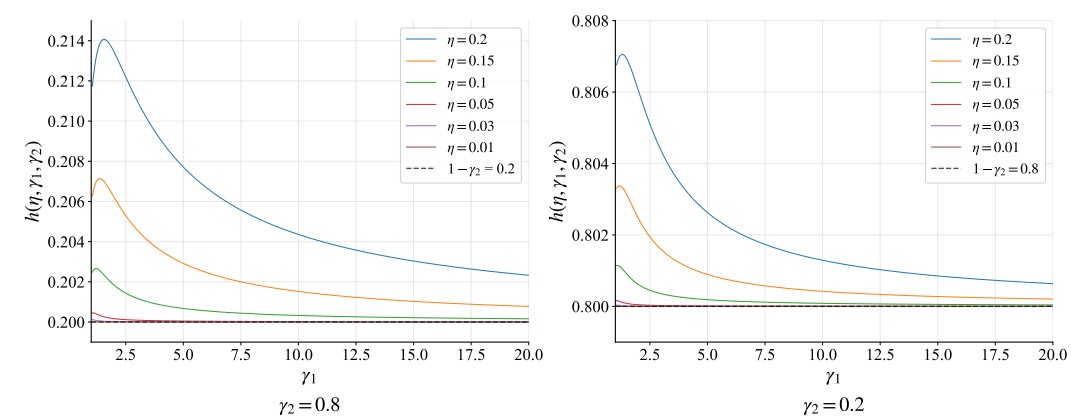

Figure 3: The function $h(\eta, \gamma_1, \gamma_2)$ versus $\gamma_1 > 1$ for different $\eta$ values with $\gamma_2 = 0.8$ and $\gamma_2 = 0.2$. When $\eta$ is small, $h(\eta, \gamma_1, \gamma_2)$ decreases almost monotonically as $\gamma_1$ increases. For larger $\eta$ (i.e., under strong regularization), $h(\eta, \gamma_1, \gamma_2)$ initially increases and then decreases monotonically with $\gamma_1$. In all cases, as $\gamma_1$ grows large, $h(\eta, \gamma_1, \gamma_2)$ converges to the dashed line $1 - \gamma_2$, as established in Corollary 4.2.

# D    OMITTED PROOFS IN SECTION 5

## D.1    PROOF OF COROLLARY 5.1

We first restate Corollary 5.1 as follows.

**Corollary 5.1.** *Let $\phi(x) = \sum x_i \log x_i$ and define the reverse cross-entropy (RCE) as $\mathrm{RCE}(y, \hat{y}) \triangleq -\sum_{i=1}^{K} \hat{y}_i \log y_i$ for any $y, \hat{y} \in \mathcal{Y}$, the following inequalities hold,*

$$\mathbb{E}\left[\mathrm{CE}\left(g(X), f_{W'}(X)\right)\right] \leq \mathbb{E}\left[\mathrm{CE}\left(g(X), f_W(X)\right)\right] - \mathbb{E}\left[\mathrm{RCE}\left(f_W(X), f_{W'}(X)\right)\right] + \mathbb{E}\left[H(f_{W'}(X))\right] + \epsilon_1,$$
$$\mathbb{E}\left[\mathrm{RCE}\left(g(X), f_{W'}(X)\right)\right] \leq \mathbb{E}\left[\mathrm{RCE}\left(g(X), f_W(X)\right)\right] - \mathbb{E}\left[\mathrm{CE}\left(f_W(X), f_{W'}(X)\right)\right] + \mathbb{E}\left[H(f_{W'}(X))\right] + \epsilon_2,$$

*where $\epsilon_1, \epsilon_2$ are defined as in Theorem 3.1.*

*Proof.* Let $\phi(x) = \sum x_i \log x_i$, then $\mathrm{D}_\phi$ becomes $\mathrm{D}_{\mathrm{KL}}$. Plugging $\mathrm{D}_{\mathrm{KL}}(y||\hat{y}) = \mathrm{CE}(y, \hat{y}) - H(y)$ into Eq. (4), we have

$$\mathbb{E}_{X,W'}\left[\mathrm{CE}\left(g(X), f_{W'}(X)\right)\right] - \mathbb{E}_X\left[H(g(X))\right]$$
$$\leq \mathbb{E}_{X,W}\left[\mathrm{CE}\left(g(X), f_W(X)\right)\right] - \mathbb{E}_X\left[H(g(X))\right]$$
$$- \mathbb{E}_{X,W',W}\left[\mathrm{RCE}\left(f_W(X), f_{W'}(X)\right)\right] + \mathbb{E}_{X,W'}\left[H(f_{W'}(X))\right] + \epsilon_1.$$

Hence,

$$\mathbb{E}_{X,W'}\left[\mathrm{CE}\left(g(X), f_{W'}(X)\right)\right] \leq \mathbb{E}_{X,W}\left[\mathrm{CE}\left(g(X), f_W(X)\right)\right]$$
$$- \mathbb{E}_{X,W',W}\left[\mathrm{RCE}\left(f_W(X), f_{W'}(X)\right)\right] + \mathbb{E}_{X,W'}\left[H(f_{W'}(X))\right] + \epsilon_1.$$

Similarly, substituting $\mathrm{D}_{\mathrm{KL}}(y||\hat{y}) = \mathrm{CE}(y, \hat{y}) - H(y)$ into Eq. (5), we have

$$\mathbb{E}_{X,W'}\left[\mathrm{RCE}\left(g(X), f_{W'}(X)\right)\right] - \mathbb{E}_{X,W'}\left[H(f_{W'}(X))\right]$$
$$\leq \mathbb{E}_{X,W}\left[\mathrm{RCE}\left(g(X), f_W(X)\right)\right] - \mathbb{E}_{X,W}\left[H(f_W(X))\right]$$
$$- \mathbb{E}_{X,W',W}\left[\mathrm{CE}\left(f_W(X), f_{W'}(X)\right)\right] + \mathbb{E}_{X,W}\left[H(f_W(X))\right] + \epsilon_2.$$

Hence,

$$\mathbb{E}_{X,W'}\left[\mathrm{RCE}\left(g(X), f_{W'}(X)\right)\right] \leq \mathbb{E}_{X,W}\left[\mathrm{RCE}\left(g(X), f_W(X)\right)\right]$$
$$- \mathbb{E}_{X,W',W}\left[\mathrm{CE}\left(f_W(X), f_{W'}(X)\right)\right] + \mathbb{E}_{X,W'}\left[H(f_{W'}(X))\right] + \epsilon_2.$$

This completes the proof. $\qquad\square$

### D.2 PROOF OF PROPOSITION 1

**Proposition 1** (Restatement). *Under the ideal conditions of Corollary 3.1 where $\epsilon_1$ and $\epsilon_2$ vanish, the performance gain in the first inequality of Corollary 5.1 coincides with Eq. (6), while the gain in the second inequality is strictly smaller than Eq. (7).*

*Proof.* If $f_{W'}(x) = \mathcal{E}\left[f_W(x)|W'\right]$ for $\forall x \in \mathcal{X}$, by Corollary 3.1 and Corollary 5.1, we have

$$
\begin{aligned}
\mathbb{E}_{X,W'}\left[\mathrm{CE}\left(g(X), f_{W'}(X)\right)\right] =& \mathbb{E}_{X,W}\left[\mathrm{CE}\left(g(X), f_W(X)\right)\right] - \mathbb{E}_{X,W',W}\left[\mathrm{RCE}\left(f_W(X), \mathcal{E}\left[f_W(X)|W',X\right]\right)\right] \\
& + \mathbb{E}_{X,W'}\left[H\left(\mathcal{E}\left[f_W(X)|W',X\right]\right)\right] \\
=& \mathbb{E}_{X,W}\left[\mathrm{CE}\left(g(X), f_W(X)\right)\right] - \mathbb{E}_{X,W,W'}\left[\mathrm{D}_{\mathrm{KL}}\left(\mathcal{E}\left[f_W(X)|W',X\right]||f_W(X)\right)\right].
\end{aligned}
$$

Thus, here the misfit term $\mathbb{E}_{X,W,W'}\left[\mathrm{D}_{\mathrm{KL}}\left(\mathcal{E}\left[f_W(X)|W',X\right]||f_W(X)\right)\right]$ matches the misfit term in Eq. (6).

In addition, if $f_{W'}(x) = \mathbb{E}\left[f_W(x)|W'\right]$ for $\forall x \in \mathcal{X}$, by Corollary 3.1 and Corollary 5.1, we have

$$
\begin{aligned}
\mathbb{E}_{X,W'}\left[\mathrm{RCE}\left(g(X), f_{W'}(X)\right)\right] =& \mathbb{E}_{X,W}\left[\mathrm{RCE}\left(g(X), f_W(X)\right)\right] - \mathbb{E}_{X,W',W}\left[\mathrm{CE}\left(f_W(X), \mathbb{E}\left[f_W(X)|W',X\right]\right)\right] \\
& + \mathbb{E}_{X,W'}\left[H(\mathbb{E}\left[f_W(X)|W',X\right])\right] \\
=& \mathbb{E}_{X,W}\left[\mathrm{RCE}\left(g(X), f_W(X)\right)\right] - \mathbb{E}_{X,W',W}\left[\mathrm{D}_{\mathrm{KL}}\left(f_W(X)||\mathbb{E}\left[f_W(X)|W',X\right]\right)\right] \\
& - \mathbb{E}_{X,W}\left[H(f_W(X))\right] + \mathbb{E}_{X,W'}\left[H(\mathbb{E}\left[f_W(X)|W',X\right])\right].
\end{aligned}
$$

Since entropy is a concave function, we have $\mathbb{E}_{X,W'}\left[H(\mathbb{E}\left[f_W(X)|W',X\right])\right] \geq \mathbb{E}_{X,W}\left[H(f_W(X))\right]$. As a result, the performance gain between $\mathbb{E}_{X,W'}\left[\mathrm{RCE}\left(g(X), f_{W'}(X)\right)\right]$ and $\mathbb{E}_{X,W}\left[\mathrm{RCE}\left(g(X), f_W(X)\right)\right]$ is larger than that in Eq. (7).

This completes the proof. $\square$

### D.3 PROOF OF PROPOSITION 2

**Proposition 2** (Restatement). *Given a data distribution $\mu = \mu_X \mu_{Y|X}$ and any $\alpha \in [0,1]$, we define a label-shifted distribution $\hat{\mu} = \mu_X \mu_{\hat{Y}|X}$ by smoothing the labels as follows: for each $(X,Y) \sim \mu$, the smoothed label is given by $\hat{Y}_j = \frac{1}{2} + \alpha\left(Y_j - \frac{1}{2}\right)$ for $\forall j \in [2]$. If RCE is used as the loss function in binary classification, then the population risk minimizer on $\hat{\mu}$ also minimizes the population risk on $\mu$.*

*Proof.* Given any model $f : \mathcal{X} \to \mathcal{Y}$, define the population risk using reverse cross-entropy as

$$
R_{rce}(f) = -\mathbb{E}_X \sum_i [f(X)]_i \log Y_i,
$$

$$
R_{rce}^\alpha(f) = -\mathbb{E}_X \sum_i [f(X)]_i \log Y_i' = -\mathbb{E}_X \sum_i [f(X)]_i \log\left(\frac{1}{K} + \alpha\left(Y_i - \frac{1}{K}\right)\right),
$$

where supervision $Y = [Y_1, \cdots, Y_K]^T$ that satisfies $\|Y\|_1 = 1$. Therefore,

$$
\begin{aligned}
R_{rce}^\alpha(f) - R_{rce}(f) &= \mathbb{E}_X \sum_i [f(X)]_i \log \underbrace{\frac{Y_i}{\frac{1}{K} + \alpha\left(Y_i - \frac{1}{K}\right)}}_{q_\alpha(Y_i)} \\
&= \mathbb{E}_X \sum_i [f(X)]_i \, q_\alpha(Y_i).
\end{aligned} \tag{45}
$$

Note that $q_\alpha(Y_i) = \log \frac{Y_i}{\frac{1}{K} + \alpha\left(Y_i - \frac{1}{K}\right)} = \log \frac{1}{\alpha}\left(1 - \frac{1-\alpha}{1-\alpha+\alpha k Y_i}\right)$ is a monotonically increasing function of $Y_i$. Let the minimizer of the ambiguous weak supervision

$$
f_\star = \arg\min_f R_{rce}^\alpha(f).
$$

There always hold $R_{rce}^\alpha(f) \geq R_{rce}^\alpha(f_\star)$. Without loss of generality, let $Y_h = 1 - \epsilon$, where $\epsilon \to 0$. For the binary classification case, i.e., $K = 2$, the inequality $R_{rce}^\alpha(f) \geq R_{rce}^\alpha(f_\star)$ means

$$-\mathbb{E}_X \sum_i [f(X)]_i \log Y_i' \geq -\mathbb{E}_X \sum_i [f_\star(X)]_i \log Y_i'$$

$$\Rightarrow \mathbb{E}_X \sum_i [f_\star(X) - f(X)]_i \log Y_i' \geq 0$$

$$\Rightarrow \mathbb{E}_X [f_\star(X) - f(X)]_h \log Y_h' + \mathbb{E}_X [f_\star(X) - f(X)]_{1-h} \log Y_{1-h}' \geq 0$$

$$\Rightarrow \mathbb{E}_X [f_\star(X) - f(X)]_h \log Y_h' - \mathbb{E}_X [f_\star(X) - f(X)]_h \log(1 - Y_h') \geq 0$$

$$\Rightarrow \mathbb{E}_X [f_\star(X) - f(X)]_h \log \frac{Y_h'}{1 - Y_h'} \geq 0$$

$$\Rightarrow \mathbb{E}_X [f_\star(X) - f(X)]_h \geq 0$$

$$\Rightarrow \mathbb{E}_X [f_\star(X) - f(X)]_{1-h} \leq 0$$

By substituting $f_\star$ into Eq. (45) and we derive an additional similar equation. Combining this new equation with the original Eq. (45) yields the below results:

$$[R_{rce}^\alpha(f) - R_{rce}(f)] - [R_{rce}^\alpha(f_\star) - R_{rce}(f_\star)]$$
$$= [R_{rce}^\alpha(f) - R_{rce}^\alpha(f_\star)] - [R_{rce}(f) - R_{rce}(f_\star)]$$
$$= \mathbb{E}_X \sum_i [f(X) - f_\star(X)]_i \, q_\alpha(Y_i)$$
$$= \mathbb{E}_X [f(X) - f_\star(X)]_h \, q_\alpha(Y_h) + \mathbb{E}_X [f(X) - f_\star(X)]_{1-h} \, q_\alpha(Y_{1-h})$$
$$= \mathbb{E}_X [f_\star(X) - f(X)]_{1-h} \, q_\alpha(Y_h) + \mathbb{E}_X [f(X) - f_\star(X)]_{1-h} \, q_\alpha(Y_{1-h}) \qquad (\|f_\star(X)\|_1 = 1)$$
$$= \mathbb{E}_X [f(X) - f_\star(X)]_{1-h} \, [q_\alpha(Y_{1-h}) - q_\alpha(Y_h)].$$

We have $Y_{1-h} \leq Y_h$, which means $q_\alpha(Y_{1-h}) \leq q_\alpha(Y_h)$. Also, $\mathbb{E}_X[f(X)]_{1-h} \geq \mathbb{E}_X[f_\star(X)]_h$. It leads to $[R_{rce}^\alpha(f) - R_{rce}^\alpha(f_\star)] - [R_{rce}(f) - R_{rce}(f_\star)] \leq 0$, i.e.,

$$0 \leq R_{rce}^\alpha(f) - R_{rce}^\alpha(f_\star) \leq R_{rce}(f) - R_{rce}(f_\star).$$

Thus, $R_{rce}(f_\star) \leq R_{rce}(f)$ holds for any $f$, which contributes to the final result

$$f_\star = \arg \min_f R_{rce}(f).$$

$$\square$$

## D.4 GRADIENT ANALYSIS

In the setting of predictive uncertainty, to further demonstrate the advantage of RCE described in Section 5, we introduce a gradient analysis for forward CE/KL and reverse CE/KL losses, which are shown in Eq. (46)-(49).

$$\frac{\partial \, \text{CE}\,(g(X), f_{W'}(X))}{\partial \, [f_{W'}(X)]_j} = -\frac{\partial \, \sum_{i=1}^2 [g(X)]_i \log [f_{W'}(X)]_i}{\partial \, [f_{W'}(X)]_j} = -\frac{[g(X)]_j}{[f_{W'}(X)]_j} + \frac{1 - [g(X)]_j}{1 - [f_{W'}(X)]_j}, \tag{46}$$

$$\frac{\partial \, \text{D}_{\text{KL}}\,(g(X) \| f_{W'}(X))}{\partial \, [f_{W'}(X)]_j} = \frac{\partial \, \text{CE}\,(g(X), f_{W'}(X)) - \partial \, H(g(X))}{\partial \, [f_{W'}(X)]_j} = \frac{\partial \, \text{CE}\,(g(X), f_{W'}(X))}{\partial \, [f_{W'}(X)]_j}, \tag{47}$$

$$\frac{\partial \, \text{RCE}\,(g(X), f_{W'}(X))}{\partial \, [f_{W'}(X)]_j} = -\frac{\partial \, \sum_{i=1}^2 [f_{W'}(X)]_i \log [g(X)]_i}{\partial \, [f_{W'}(X)]_j} = \log \frac{1 - [g(X)]_j}{[g(X)]_j}. \tag{48}$$

$$\frac{\partial \, \text{D}_{\text{KL}}\,(f_{W'}(X) \| g(X))}{\partial \, [f_{W'}(X)]_j} = \log \frac{1 - [g(X)]_j}{[g(X)]_j} - \log \frac{1 - [f_{W'}(X)]_j}{[f_{W'}(X)]_j} \tag{49}$$

As the student prediction $f_{W'}(x)$ approaches the supervision $g(x)$, the gradients for forward CE/KL and reverse KL diminish to zero. In contrast, the reverse CE gradient persists, maintaining non-vanishing values that facilitate more robust learning under uncertain supervision. It will be empirically validated in Appendix E.3.

# E EXPERIMENTAL DETAILS AND ADDITIONAL RESULTS

## E.1 EXPERIMENTAL DETAILS

**Dataset Processing** For standard NLP tasks (SciQ (Welbl et al., 2017), Amazon Polarity(McAuley & Leskovec, 2013) and Twitter Sentiment), we convert the original questions and candidate answers $Q, A_1, ..., A_k$ into multiple question-answer pairs $(Q, A_i)$, where correct answers are labeled as positive and incorrect ones as negative. For reward modeling tasks (CAI-Harmless (Bai et al., 2022b), HH-RLHF (Bai et al., 2022a)), we directly pair the chosen and rejected responses while maintaining their original preference labels. All datasets maintain class balance and are partitioned into three subsets: $S$, $S'$, and $S_{test}$, designed for weak model training, strong model training, and final performance evaluation, respectively.

**Model Architecture** For standard NLP tasks (SciQ, Amazon Polarity, Twitter Sentiment), with experimental setup of (Burns et al., 2024), we add a two-dimensional linear projection layer atop the pretrained model's final representations, followed by a Softmax function to obtain the final prediction probabilities. For reward modeling tasks (CAI-Harmless, HH-RLHF), with experimental settings referencing (Yao et al., 2025a;b), we instead use a single-output linear layer with a Sigmoid activation to map the output to a probability in [0, 1], indicating whether the response meets the harmlessness or helpfulness criteria.

**Training Configurations** We implement early stopping to prevent overfitting, using minimal training epochs for efficiency (Burns et al., 2024). Standard NLP tasks train for 2 epochs while reward modeling tasks use only 1 epoch. The base learning rate is set to $1 \times 10^{-5}$, except for GPT2 and GPT2-Medium models on SciQ and Amazon Polarity datasets which employ $5 \times 10^{-5}$. Batch sizes are configured as 32 for NLP tasks and 16 for reward modeling tasks. All experiments run on 4 NVIDIA vGPUs (32GB) with at least three independent random seeds for reproducibility. Complete configurations are summarized in Table 2. Notably, we adopt a full fine-tuning strategy during training without freezing any pretrained parameters, allowing the model to fully adapt to downstream tasks.

Table 2: Training Configurations on Different Datasets.

| Dataset | $|S|$ | $|S'|$ | $|S_{test}|$ | Max Seq Len | Epochs | Batch Size |
|---|---|---|---|---|---|---|
| SciQ | 5839 | 5838 | 1000 | 1024 | 2 | 32 |
| Amazon Polarity | 5000 | 5000 | 1000 | 1024 | 2 | 32 |
| Twitter Sentiment | 7000 | 7000 | 1000 | 1024 | 2 | 32 |
| CAI-Harmless | 4000 | 4000 | 4000 | 512 | 1 | 16 |
| HH-RLHF | 4000 | 4000 | 4000 | 512 | 1 | 16 |

**Loss Fuctions** The Confidence-Adaptive Cross Entropy (CACE) loss dynamically combines reverse cross-entropy (RCE) and standard cross-entropy (CE) based on teacher confidence levels. Formally, it is defined as:

$$\mathcal{L}_{\text{CACE}}(y, \hat{y}) = \mathbb{I}(y, c) \cdot \text{RCE}(y, \hat{y}) + \big(1 - \mathbb{I}(y, c)\big) \cdot \text{CE}(y, \hat{y}), \tag{50}$$

where $\hat{y}$ represents the student's prediction and $\mathbb{I}(y, c)$ is an indicator function that activates when the teacher's soft label $y$ has confidence below threshold $c$. For our binary classification tasks, we quantify confidence as $|y - 0.5|$, where $y \in (0, 1)$ is the teacher's soft label. The threshold $c$ is determined adaptively before W2S training by selecting a quantile such that exactly $\eta\%$, of the samples are viewed as low-confidence, i.e., activate $\mathbb{I}(y, c)$. In our experiments, $\eta \in \{5, 10, 20, 30\}$.

The Symmetric Cross Entropy Loss (SL) (Wang et al., 2019) is designed to handle noisy labels by balancing reverse cross-entropy (RCE) and standard cross-entropy (CE). It is formally defined as:

$$\mathcal{L}_{\text{SL}}(y, \hat{y}) = \lambda_1 \cdot \text{RCE}(y, \hat{y}) + \lambda_2 \cdot \text{CE}(y, \hat{y}), \tag{51}$$

where $\hat{y}$ represents the student's prediction, and $\lambda_1$ and $\lambda_2$ are weighting factors that adjust the contributions of RCE and CE respectively. In our experiments, we evaluate the performance of SL under different configurations of $(\lambda_1, \lambda_2) \in \{(1.0, 0.5), (1.0, 0.1), (1.0, 1.0), (0.5, 1.0), (0.1, 1.0)\}$.

The Auxiliary Confidence Loss (AUX) (Burns et al., 2024) is primarily aimed at enhancing the confidence of the student model's predictions through regularization. It is defined as:

$$\mathcal{L}_{\text{AUX}} = \beta\text{CE}(f_w(x), f_{w'}(x)) + (1 - \beta)\text{CE}(\hat{f}_{w'}(x), f_{w'}(x)), \tag{52}$$

where $\text{CE}(\cdot, \cdot)$ denotes the cross-entropy loss between two prediction distributions. Here, $f_w(x)$ represents the weak label prediction from one model, $f_{w'}(x)$ is the prediction from another weakly supervised model, and $\hat{f}_{w'}(x)$ is the hardened prediction of the student model, which becomes a one-hot vector when $f_{w'}(x)$ exceeds a certain confidence threshold. Following (Burns et al., 2024), the confidence threshold $t$ is adjusted so that exactly half of the samples in a batch have $f(x) > t$, ensuring that only sufficiently confident predictions are considered for hardening. Additionally, the weight parameter $\beta$ undergoes a linear warm-up phase, increasing from 0 to its maximum value $\beta_{\max}$ during the initial training period. For our experiments, we apply this warm-up strategy separately over the first 20%, 50%, and 100% of the training data.

Notably, the results for CACE, SL, and AUX shown in Table 1 are the best outcomes selected from the above-mentioned hyperparameter settings.

### E.2 Bias and Variance Estimation in W2SG

Following (Yang et al., 2020), we provide the bias-variance decomposition for cross-entropy (CE) in our work, which is formally given by:

$$\underbrace{\mathbb{E}_X\mathbb{E}_\theta[\text{CE}\left(g(X), f_\theta(X)\right)]}_{\text{Population Risk}} = \underbrace{\mathbb{E}_X[\text{D}_{\text{KL}}\left(g(X)||\mathcal{E}_\theta[f_\theta(X)]\right)]}_{\text{Bias}^2} + \underbrace{\mathbb{E}_X\mathbb{E}_\theta[\text{D}_{\text{KL}}\left(\mathcal{E}_\theta[f_\theta(X)]||f_\theta(X)\right)]}_{\text{Variance}},$$
$$\tag{53}$$

where the model parameter $\theta \in \{W, W'\}$, and the intrinsic randomness of $\theta$ stems from several sources, such as data sampling, model initialization and optimization. Here, $X$ denotes a sample from the test set. $\mathcal{E}_\theta[f_\theta(X)]$ is the average of log-probability after normalization.

To accurately estimate $\mathcal{E}_\theta[f_\theta(X)]$ in Eq. (53), we train multiple independent teacher-student pairs on distinct subsets sampled from large-scale datasets and average their performance. Specifically, we construct five disjoint 10000-sample subsets from Amazon Polarity (with equal halves allocated to weak model training and W2S training), training five teacher-student pairs across four random seeds. Similarly, we extract three 14000-sample subsets from Twitter Sentiment, conducting experiments across five random seeds. The complete implementation details are provided in Algorithm 1, and the results are shown as solid lines in Figure 1 and Figure 4.

To validate Corollary 3.1 – which states that W2SG emerges when the student's prediction matches its (dual) posterior mean teacher – we implement a probability-based ensemble (Dietterich, 2000; Zhou, 2025) of multiple weak teachers for student supervision. Specifically, we compute the dual expectation $\mathcal{E}[f_W(X)]$ over weak teachers' predictions to approximate the teacher's conditional expectation $\mathcal{E}[f_W(X)|W', X]$ in Eq. (6). When using CE loss for teacher training, $\mathcal{E}[f_W(X)]$ is computed as:

$$\mathcal{E}[f_W(X)] = \frac{\exp\left(\mathbb{E}_W[\log f_W(X)]\right)}{\sum_{i=1}^{K} \exp\left(\mathbb{E}_W \log\left[f_W(X)]_i\right)\right)}, \tag{54}$$

where $[f_W(X)]_i$ denotes the probability of sample $X$ belonging to class $i$ as predicted by the teacher model, and $K$ represents the total number of classes. Note that in this context, $X$ represents a fixed sample from the W2S training set, and the expectation is taken only over the weak teachers. We use these ensemble predictions as training data to supervise student models.

The results for the Amazon Polarity dataset are presented as dashed lines in Figure 1, and the results for the Twitter Sentiment dataset are shown in Figure 4. These results indicate that greater model complexity generally leads to larger performance gains, demonstrating that W2SG is more likely to emerge when the student model is more capable, which is consistent with the insights in Theorem 4.1 and Remark 4.1. Moreover, when the size gap between teacher and student models is too large, the misfit term tends to vanish monotonically, limiting performance improvement. For example, in Figure 1, the loss reduction of Qwen-14B is smaller than that of Qwen-7B. This suggests the existence of a subtle tradeoff that warrants further investigation.

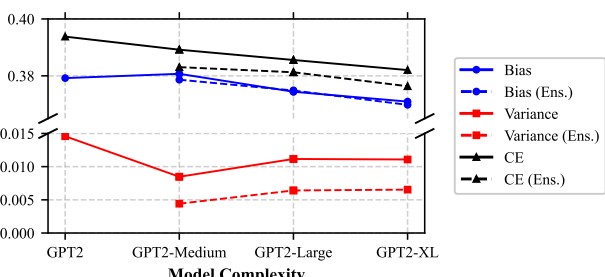

Figure 4: Bias and variance estimation on Twitter Sentiment. GPT-2 is trained as the teacher and used to supervise the three strong students' model traing. "CE" represents cross-entropy test loss. "Ens." represents student performance supervised by expected teacher, approximated via weak teachers ensemble in dual space.

---

**Algorithm 1** Estimate W2SG Bias and Variance

**Input:** Test point $\mathbf{x}$, weak model training set $S$ and strong model training set $S'$
**for** $i = 1$ **to** $k$ **do**
    Split the $(S, S')$ into $N$ pairs: $(S_1^{(i)}, S_1'^{(i)}), \ldots, (S_N^{(i)}, S_N'^{(i)})$
    **for** $j = 1$ **to** $N$ **do**
        Train the weak model $f_w$ using $S_j^{(i)}$
        Evaluate the weak model at $\mathbf{x}$; call the result $\pi_j^{(i)}$
        Generate pseudo-labels for $S_j'^{(i)}$ using $f_w$
        Train the strong model using $S_j'^{(i)}$
        Evaluate the W2S model at $\mathbf{x}$; call the result $\pi_j'^{(i)}$;
    **end for**
**end for**
Compute $\widehat{\pi} = \exp\left\{ \frac{1}{N\cdot k} \sum_{ij} \log\left(\pi_j^{(i)}\right) \right\}$, $\widehat{\pi}' = \exp\left\{ \frac{1}{N\cdot k} \sum_{ij} \log\left(\pi_j'^{(i)}\right) \right\}$
(*using element-wise log and exp; $\widehat{\pi}$ estimates $\bar{\pi}$*).
Normalize $\widehat{\pi}$ to get a probability distribution.
Compute the variance $\frac{1}{N\cdot k} \sum_{ij} D_{\text{KL}}\left(\widehat{\pi} || \pi_j^{(i)}\right)$, $\frac{1}{N\cdot k} \sum_{ij} D_{\text{KL}}\left(\widehat{\pi}' || \pi_j'^{(i)}\right)$.
Compute the bias $\frac{1}{N\cdot k} \sum_{ij} D_{\text{KL}}\left(\pi_0 || \widehat{\pi}\right)$, $\frac{1}{N\cdot k} \sum_{ij} D_{\text{KL}}\left(\pi_0 || \widehat{\pi}'\right)$.
where $\pi_0(\mathbf{x})$ be the one-hot encoding of the ground-truth label.

---

### E.3 ADDITIONAL EMPIRICAL VALIDATION OF RCE'S ROBUSTNESS TO PREDICTIVE UNCERTAINTY

**Gradient Dynamics in CE and RCE** In addition to the Amazon Polarity and HH-RLHF datasets presented in Figure 2, we further compare the sensitivity of CE and RCE losses to predictive uncertainty on SciQ and CAI-Harmless datasets. We observe that models trained with RCE tend to move farther in the parameter space. As shown in Figure 2 and Figure 5, nearly all RCE Distance values (purple line) are higher than CE Distance values (green line). When $\alpha = 0$, $\mathcal{L}_{\text{RCE}} = \log K$ becomes constant for $K$-classification, resulting in zero gradient and thus RCE Distance equals zero. We further track the training dynamics via gradient norm and gradient direction variance (GDV) (Liu et al., 2023), where the GDV metric quantifies the directional consistency of mini-batch gradients during training.

$$\text{GDV} = \frac{1}{|G| \cdot (|G| - 1)} \sum_{g_i, g_j \in G, i \neq j} \left( 1 - \frac{\langle g_i, g_j \rangle}{\|g_i\|_2 \cdot \|g_j\|_2} \right), \tag{55}$$

where $G$ denotes a set of mini-batch gradients. As shown in Figure 6, although RCE has smaller gradient norms, its GDV remains low, suggesting more consistent update directions. Conversely, CE exhibits higher GDV, leading to more "meandering" updates and causing the model to "wander" around the initial point.

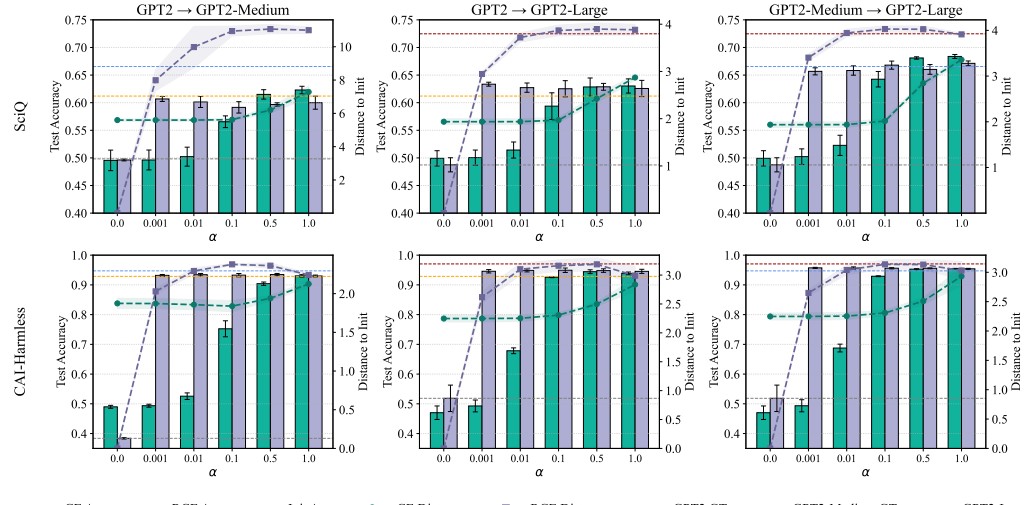

Figure 5: Performance of the GPT2 series models across SciQ and CAI-Harmless under varying $\alpha$ values, comparing CE and RCE losses. "GPT2 $\rightarrow$ GPT2-Medium" denotes GPT2 supervising GPT2-Medium. Left y-axis shows test accuracy, corresponding to the bar plots for CE Acc and RCE Acc. Right y-axis illustrates the $L_2$ norm between the fine-tuned and initial models, represented by the line plots for CE Distance and RCE Distance. The cases $\alpha = 0$ and $\alpha = 1$ represent uniform and unshifted pseudo-labels, respectively. GT denotes the test accuracy achieved by training with ground truth labels. All experiments are repeated three times.

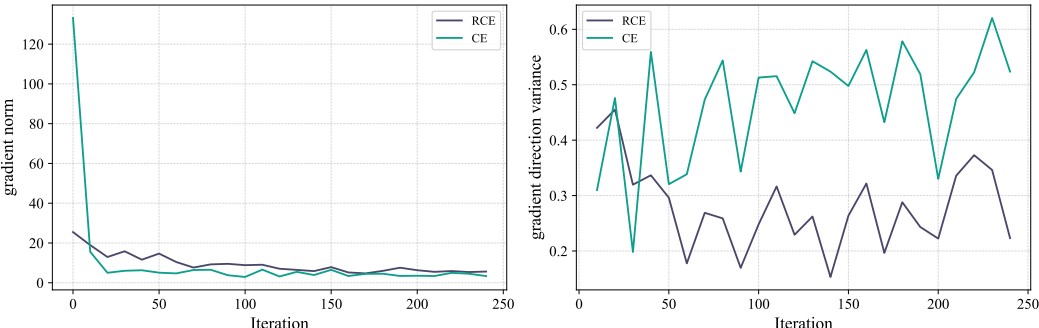

Figure 6: Gradient norm (left) and gradient direction variance (GDV, right) on the CAI-Harmless dataset, where GPT2-Medium is supervised by GPT2. RCE demonstrates lower GDV than CE, indicating stronger gradient consistency.

**Comparison of CE, RCE, KL and RKL in W2SG**    Prior works (Yao et al., 2025a) investigated the mode-seeking behavior (Minka et al., 2005) of reverse KL divergence (RKL) in W2SG, establishing its advantages. We further conduct a systematic comparison of CE, RCE, KL, and RKL when applying the label-proportional smoothing strategy as outlined in Proposition 2. As shown in Fig. 7, RCE demonstrates significantly stronger robustness to predictive uncertainty compared to the other three losses on both CAI-Harmless and HH-RLHF datasets. This empirically validates Proposition 2 and Appendix D.4 regarding RCE's gradient stability advantages, despite RKL differing from RCE by only an entropy term.

**Comparison of CE and RCE in Knowledge Distillation**    We also investigate RCE under standard knowledge distillation (Hinton et al., 2015) settings, where a high-complexity teacher model supervises a low-complexity student model. Specifically, we consider two setups: (1) GPT2-Medium serves as the teacher, providing pseudo-labels to supervise GPT2; and (2) GPT2-Large supervises both GPT2 and GPT2-Medium. Results are summarized in Figure 8. Consistent with our findings in W2SG, RCE maintains significantly higher accuracy on samples with low-confidence predictions compared to CE. Models trained with RCE exhibit more consistent gradient directions and achieve larger parameter updates during training, indicating better learning stability and efficiency.

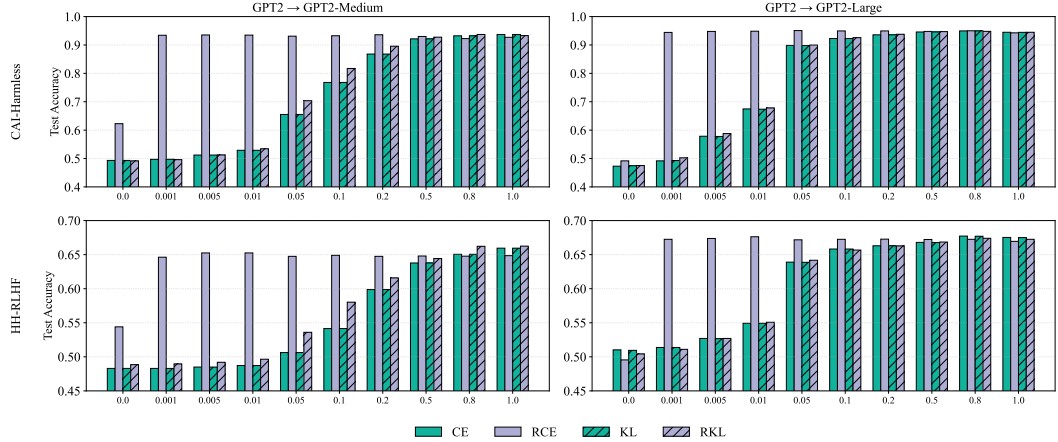

Figure 7: Performance of the GPT2 series models across CAI-Harmless and HH-RLHF under varying $\alpha$ values, comparing CE, RCE, KL and RKL losses. "GPT2-Medium $\rightarrow$ GPT2" denotes GPT2-Medium supervising GPT2. The cases $\alpha = 0$ and $\alpha = 1$ represent uniform and unshifted pseudo-labels, respectively.

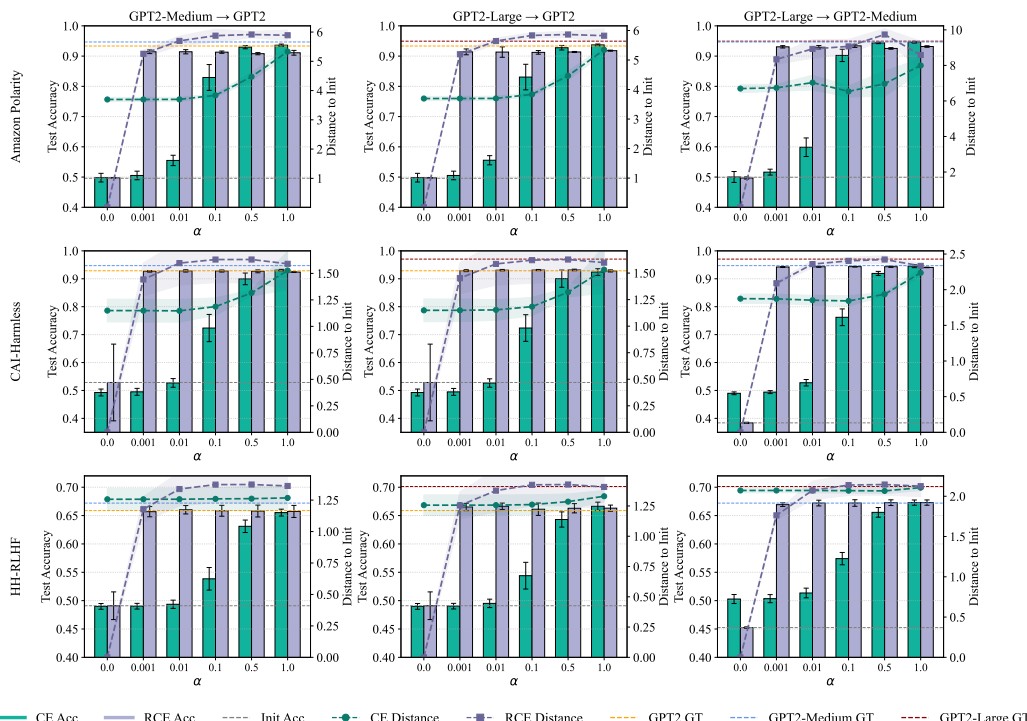

Figure 8: Performance of the GPT2 series models across Amazon Polarity, CAI-Harmless and HH-RLHF under varying $\alpha$ values, comparing CE and RCE losses. "GPT2-Medium $\rightarrow$ GPT2" denotes GPT2-Medium supervising GPT2. Left y-axis shows test accuracy, corresponding to the bar plots for CE Acc and RCE Acc. Right y-axis illustrates the $L_2$ norm between the fine-tuned and initial models, represented by the line plots for CE Distance and RCE Distance. The cases $\alpha = 0$ and $\alpha = 1$ represent uniform and unshifted pseudo-labels, respectively. GT denotes the test accuracy achieved by training with ground truth labels. All experiments are repeated three times.

**Validation on Larger Models**  In addition to the GPT-2 series, we validated the comparative experiments of RCE and CE under the W2SG and Knowledge Distillation settings using the Qwen series (Qwen-0.5B, Qwen-3B, Qwen-7B), as shown in Figure 9. Our results indicate that on more complex models, RCE continues to demonstrate its advantage in situations with low prediction confidence.

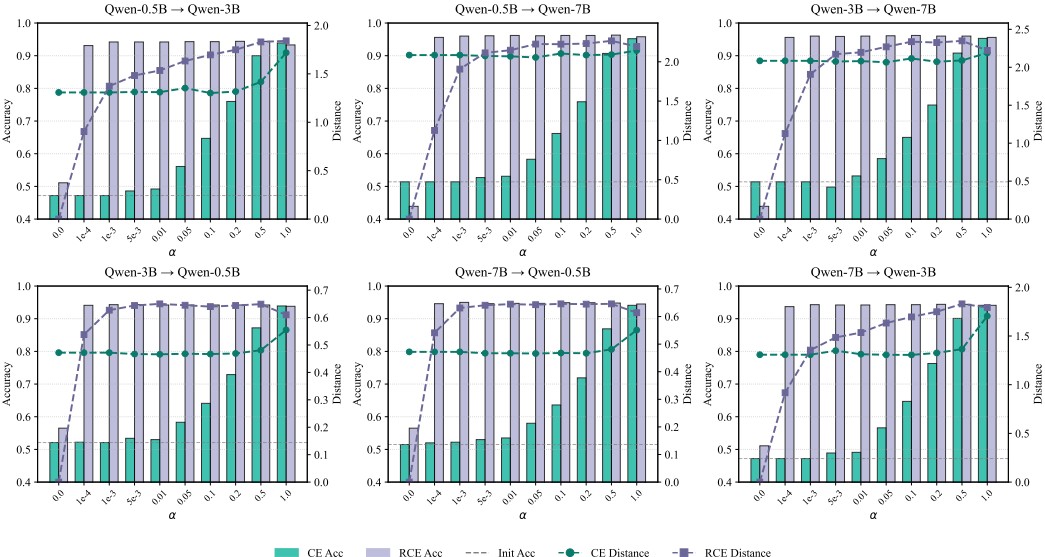

Figure 9: Performance of the Qwen series models on CAI-Harmless under varying $\alpha$ values, comparing CE and RCE losses. "Qwen-0.5B $\to$ Qwen-3B" denotes Qwen-0.5B supervising Qwen-3B. Left y-axis shows test accuracy, corresponding to the bar plots for CE Acc and RCE Acc. Right y-axis illustrates the $L_2$ norm between the fine-tuned and initial models, represented by the line plots for CE Distance and RCE Distance. The cases $\alpha = 0$ and $\alpha = 1$ represent uniform and unshifted pseudo-labels, respectively.

## F ADDITIONAL EXPERIMENTAL RESULTS OF CACE AND SL

Table 3: Test accuracy (%) of five loss functions on Amazon Polarity and ImageNet datasets. The optimal and suboptimal results are marked in **bold** and underline, respectively.

| Dataset | Model | CE | RCE | AUX | CACE | SL |
|---------|-------|-----|-----|-----|------|-----|
| Amazon Polarity | Qwen-1.8B $\to$ Qwen-7B | 95.50 | 96.30 | 95.80 | 96.50 | **96.60** |
| | Qwen-1.8B $\to$ Qwen-14B | 95.90 | 96.30 | 96.50 | **96.70** | 96.50 |
| | Qwen-7B $\to$ Qwen-14B | 96.40 | 96.40 | 96.90 | **97.10** | 96.80 |
| ImageNet | AlexNet $\to$ ResNet-50 (DINO) | 61.70 | 44.67 | 61.69 | 61.70 | **61.72** |
| | AlexNet $\to$ ViT-B/8 (DINO) | 66.68 | 54.63 | 68.06 | **70.03** | 68.42 |

## G LLM USAGE STATEMENT

In the preparation of this manuscript, we utilized Large Language Models (LLMs), specifically GPT-4o and Gemini 2.5, as writing assistants. The use of these models was limited to language polishing, grammar correction, and improving the overall readability of the text. All suggestions from the LLMs were carefully reviewed, edited, and approved by the authors, who take full responsibility for the final content of this paper.

