# OpenReview forum: "On the Emergence of Weak-to-Strong Generalization: A Bias-Variance Perspective"
_ICLR.cc/2026/Conference — Submitted to ICLR 2026_

### Official Review · Reviewer_WpDr · 2025-10-29

**Soundness:** 2
**Presentation:** 1
**Contribution:** 2
**Rating:** 2
**Confidence:** 2

**Summary:**

This paper studies the weak to strong generalization phenomena, where a weak teacher that provides labels to a strong student, can be outperformed by the student.

**Strengths:**

- Extensive theory

- The theoretical contribution is clear: they can remove the convexity assumption, and they show the insight that making the student more certain makes weak to strong generalization stronger

**Weaknesses:**

- I think this paper in principle has a good contribution, but its presentation is lacking. Its unclear what is the main insight / main point.

- I find the notation confusing. Sometimes the student is indicated by accents, and other times by _s. It would be nice if this is consistent. Especially f_w' throws me off; is this now "weak" and why does it have an accent? Ideally: "w" for weak and "s" for strong consistently.

- The theory seems rather "disconnected" from the experiments.

- The paper feels unstructured or unorganized. New methods (CACE, SL) are introduced in the results section, and the figures are too dense; they contain lots of information, making it unclear what is the main point.

**Questions:**

1. The conditions of Theorem 2 seem to not agree with your setting, because in your setting the strong model has a larger hypothesis class. But in this theorem, they are both sitting on top of the same backbone; so aren't this models then in the same class? Why is one weak and the other strong here?

2. In Example 1; aren't both models equally strong? They are both linear right?

3. In the figures: Why are there CE and RCE Distances here? What are they? Why are they here?

4. I felt that the Bias and Variance in the Experiments was rather disconnected from the theory. Is it possible to more clearly how they are connected? Its pretty much lost on me - it seems Appendix E.2 is defining bias and variance; but this seems different than the main terms of your theory section?

5. Were the experiments carried out multiple times and averaged? I am missing confidence intervals, and I wonder about the significance of the findings. How were the hyperparameters such as $c$ tuned?

6. Removing the convexity assumption; what did this now yield compared to other SOTA analyses?

---

> ### Author Response · Authors · 2025-11-21
> **Response to Reviewer WpDr (Part 1/3)**
>
> We thank you sincerely for your valuable feedback on our paper. Our responses follow.
>
> >W2: I find the notation confusing. Sometimes the student is indicated by accents, and other times by _s. It would be nice if this is consistent. Especially f_w' throws me off; is this now "weak" and why does it have an accent? Ideally: "w" for weak and "s" for strong consistently.
>
> **Response:** We apologize if this necessary choice caused any inconvenience in reading the paper.
>
> For the hypothesis-class dimensions, we indeed use accents to distinguish between $d_w$ and $d_s$, where $w$ denotes "weak" and $s$ denotes "strong".
>
> However, for the weak and strong models‘ parameters, we follow the common convention of using $w$ (or $W$) to denote model parameters (or their associated random variables). For this reason, we use $f_W$ for the weak model and $f_{W'}$ for the strong model, where the accent does not denote "weak".
>
> We also avoid using $f_S$ for the strong model because $S$ already denotes the dataset, and reusing it would risk confusion with the strong model’s random variable.
>
>
> >W3: The theory seems rather "disconnected" from the experiments.
>
> **Response:** We respectfully disagree that the theory is "disconnected" from the experiments, and we clarify the connection below.
>
> First, Thm. 3.1 (Cor. 3.1) establishes a sufficient condition for the emergence of W2SG: the student matches its posterior mean teacher. Our experiments directly validate this prediction. We approximate the expected teacher using model ensembles and use this ensemble to supervise the student (the dashed curves in Fig. 1), while supervision from a single teacher corresponds to the solid curves. The results show that when the student approaches its posterior mean teacher, its performance surpasses that of mimicking an individual teacher, indicating that W2SG is more likely to occur. Moreover, across different architectures, increasing the student model size steadily improves performance, which is consistent with Thm. 4.1 showing that larger student capacity promotes W2SG.
>
> Second, in Section 5, we theoretically show that RCE loss is significantly less sensitive to the teacher's predictive uncertainty. We empirically verify this by smoothing the teacher's pseudo-labels and comparing CE and RCE under varying confidence levels. RCE remains robust even when the pseudo-labels are highly uncertain, whereas CE does not, directly supporting our theoretical analysis.
>
> Third, based on these theoretical analyses and empirical observations, we propose the CACE loss, which combines the strengths of CE and RCE to adapt to the varying confidence levels of pseudo-labels in W2S training, and it demonstrates superior performance compared with other methods.
>
> Overall, the first two experiments serve as empirical validations of our theoretical results, while the third is an algorithmic design directly inspired by these theoretical insights, showing that the theory and experiments are closely connected.
>
> >W1: I think this paper in principle has a good contribution, but its presentation is lacking. Its unclear what is the main insight / main point.
> >
> >W4: The paper feels unstructured or unorganized. New methods (CACE, SL) are introduced in the results section, and the figures are too dense; they contain lots of information, making it unclear what is the main point.
>
> **Response:** We thank the reviewer for recognizing the contribution of our work. We have revised the manuscript, particularly the introduction, to make the main point more explicit and easier for readers to follow.
>
> Here we summarize the structure and main insights of the paper:
>
> 1. We establish **W2SG inequalities without convexity assumptions** based on a generalized bias–variance decomposition of Bregman divergence, and further derive a sufficient condition for W2SG to occur—namely, that the **student model aligns with its posterior mean teacher**. Under symmetric Bregman divergence (i.e., squared loss), this implies that **a sufficiently large student model ensures W2SG**.
> 2. Extending to asymmetric Bregman divergence, we show that **increasing the student’s predictive confidence (reducing entropy) promotes W2SG**. Furthermore, we theoretically demonstrate that **RCE is robust to the teacher's predictive uncertainty**.
> 3. We empirically verify the theoretical sufficient conditions for W2SG and the robustness of RCE, and **propose CACE loss to effectively leverage information from the teacher's label confidence**.
>
> Overall, while the paper presents a rich set of results, we aim to connect the theory coherently by moving from general W2SG inequalities under Bregman divergence to concrete symmetric and asymmetric cases. We have carefully restructured our introduction section (Lines 83–101) based on this logic.
>
> Regarding the comment that "the figures are too dense", we have also adjusted the layouts to improve readability.

---

> ### Author Response · Authors · 2025-11-21
> **Response to Reviewer WpDr (Part 2/3)**
>
> >Q1: The conditions of Theorem 2 seem to not agree with your setting, because in your setting the strong model has a larger hypothesis class. But in this theorem, they are both sitting on top of the same backbone; so aren't this models then in the same class? Why is one weak and the other strong here?
>
> **Response:** Yes, Thm. 2.1 is different from our setting. It is presented as an informal statement of Mulgund & Pabbaraju (2025) to facilitate comparison with our results. The formal version appears in Appendix A.
>
> In their theorem, the strong model’s fine-tuning function is restricted to a convex set—a strong assumption that does not reflect practical scenarios. In contrast, our W2SG inequalities do not require such assumptions.
>
> Moreover, in their setup, the strong model’s fine-tuning functions sit on top of different backbones, with the strong model's backbone being deeper, so the models belong to different classes. The theorem considers a fixed backbone and only fine-tunes $f_{W'}$.
>
> We have revised the statement of Theorem 2.1 to prevent potential misunderstandings.
>
> >Q2: In Example 1; aren't both models equally strong? They are both linear right?
>
> **Response:** No, in the context of W2SG, "strong" refers to a larger model size or more parameters. Both the weak model $f_w$ and the strong model $f_{w'}$ are linear, but $f_{w'}$ clearly has more parameters, making it the stronger model.
>
> >Q3: In the figures: Why are there CE and RCE Distances here? What are they? Why are they here?
>
> **Response:** CE / RCE Distances refer to the $L_2$ norm between the fine-tuned and the initial model parameters, computed when using CE or RCE as the training loss.
>
> We empirically included these distances to provide insight from an optimization perspective, as detailed in Appendix E.3. By tracking the training dynamics via gradient norm and gradient direction variance (GDV), we observe that RCE as a training objective produces more consistent gradient update directions, leading the model to move farther from its initialization. This observation aligns with the gradient analysis in Appendix D.4.
>
> We acknowledge that presenting CE and RCE distances in Figure 2 of the main text may cause confusion, and we have added a brief clarification in the text, including the caption of Figure 2 and Lines 472–475.
>
>
> >Q4: I felt that the Bias and Variance in the Experiments was rather disconnected from the theory. Is it possible to more clearly how they are connected? Its pretty much lost on me - it seems Appendix E.2 is defining bias and variance; but this seems different than the main terms of your theory section?
>
> **Response:** We believe there may be a minor misunderstanding here. The Bias and Variance in the Experiments is in fact tightly connected to our theoretical results.
>
> As we clarified in our response to Weakness 3, the experiments in Figure 1 are primarily designed to validate Thm. 3.1 and Thm. 4.1, which characterize (i) how the student approaches the posterior mean teacher and (ii) how increasing the student model size facilitates the emergence of W2SG. We perform a bias–variance decomposition (Eq. (53) in Appendix E.2) of the cross-entropy loss in order to further investigate **which of the two quantities—bias or variance—dominantly contributes to these improvements**.
>
> As discussed in the main experimental analysis, we observe that (i) the student approaching the posterior mean teacher is mainly driven by a reduction in variance, whereas (ii) scaling up student capacity primarily reduces bias.
>
> Correspondingly, in the theoretical proofs, we rely on the conditional bias–variance decomposition (Eq. (11) in Appendix B.1). Although Eq. (53) in Appendix E.2 takes expectations differently, it is simply the empirical counterpart of the same concept.

---

> ### Author Response · Authors · 2025-11-21
> **Response to Reviewer WpDr (Part 3/3)**
>
> >Q5: Were the experiments carried out multiple times and averaged? I am missing confidence intervals, and I wonder about the significance of the findings. How were the hyperparameters such as $c$ tuned?
>
> **Response:** Yes. All experiments were conducted with three independent random seeds, and we report the standard deviation across runs (e.g., in Figure 2). The observed trends are consistent across seeds, and the error bars are small relative to the effect sizes, indicating that the findings are robust.
>
> For the hyperparameter $c$, we follow the procedure described in Appendix E.1: the threshold is determined via a quantile-based selection of $\eta$, where $\eta$ specifies the percentage of samples treated as low-confidence. We evaluate several quantile levels $\eta \in \\{ 5, 10, 20, 30 \\} $ and the reported results use the best-performing setting among these values.  We found $\eta = 30$ generally performs best on CAI-Harmless and HH-RLHF.
>
> >Q6: Removing the convexity assumption; what did this now yield compared to other SOTA analyses?
>
> **Response:** We are not entirely sure what specific SOTA theoretical analyses the reviewer is referring to. To the best of our knowledge, our work advances the line of misfit-based analyses in a substantive way.
>
> Beyond misfit-based analyses, several theoretical works rely on additional structural assumptions. For example, Lang et al. (2024) require specific data-neighborhood conditions that enable error correction through adversarial robustness, and Dong et al. (2025) assume that the pretrained features have low intrinsic dimensionality. Our analysis does not rely on such data- or representation-level assumptions,  making it applicable to a broader class of student–teacher settings.
>
> In addition, our framework handles both the squared loss and the cross-entropy loss, while many prior theoretical works do not address both.
>
> If the reviewer has a more concrete definition of SOTA analyses, we are open to further discussion.

---

> ### Author Response · Authors · 2025-11-28
>
> Dear Reviewer WpDr,
>
> Thank you for your time and effort in reviewing our paper. As the discussion period is approaching its end, we wish to confirm whether our responses have adequately addressed your concerns. We would greatly appreciate it if you could re-evaluate our paper based on our rebuttal.
>
> Thank you again for your consideration and support.
>
> Best regards, Paper 15511 Authors

---

### Official Review · Reviewer_GSgt · 2025-10-30

**Soundness:** 2
**Presentation:** 2
**Contribution:** 1
**Rating:** 2
**Confidence:** 4

**Summary:**

The paper develops a misfit-based theory of W2SG for broad Bregman losses without assuming convexity of the student hypothesis class. The central result is an expected risk inequality (over data and model draws) that upper bounds the student’s population risk by the teacher’s risk minus an expected misfit term, up to residuals arising from teacher–student dependence. This is obtained via a generalized bias–variance decomposition for Bregman divergences, thereby avoiding the generalized Pythagorean projection used in prior works. From this lens, the paper argues that W2SG is facilitated when the student aligns to a posterior-mean teacher, proposes to ensemble weak teachers to approximate that mean, and analyzes a random-features ridge example where, as the student grows, the expected misfit shrinks and the student converges (in expectation) to the posterior-mean teacher while maintaining a positive misfit gap. Finally, for classification, the paper compares cross-entropy (CE) vs. reverse cross-entropy (RCE), deriving misfit-gain-based generalization bounds, showing the robustness of RCE to uncertain teacher predictions. Experiments empirically visualize the bias-variance decomposition through bias/variance estimates and demonstrate the comparative performance of CE vs. RCE in W2SG settings, which further motivates a new objective, named confidence-adaptive cross entropy (CACE) loss, that interpolates between CE and RCE based on teacher confidence.

**Strengths:**

1. The topic of W2SG is timely and relevant; the connection with and distinction from prior works, especially Charikar et al., 2024 and Mulgund & Pabbaraju, 2025 are clearly articulated.
2. Moving from the convex projection arguments in prior works to the expected Bregman bias–variance analysis yields misfit-gain inequalities for non-convex students without convexity or linear head restrictions, addressing a key technical limitation in Charikar et al., 2024 and Mulgund & Pabbaraju, 2025.
3. Empirically, the bias-variance estimation protocol and the ensemble-based approximation of the dual expectation are thoughtful.

**Weaknesses:**

1. While the detailed discussion on the relation with prior works and the technical novelties upfront in the introduction (lines 46-82) can provide good motivations for this work for domain experts (as mentioned in Strengths), it could be overwhelming for general audience in the community. I would suggest reorganizing the introduction carefully, e.g., partitioning the technical novelties in lines 60-82 into bullet points, each summarized by a concise, intuitive title explaining the effect/benefit.
2. From a high-level perspective, while I acknowledge the technical novelties in the main theoretical results, I feel that the significance of the improvements brought by these novelties is not sufficient to meet the bar of ICLR. In particular:
    - Mulgund & Pabbaraju, 2025 already extend misfit-gain to general Bregman losses and propose convex combinations of logistic heads for classification; this paper avoids convexity by going to the expected risk over both data and models. While the main results in Theorem 3.1 relaxes the key limitation of the convexity assumption in prior works, as a trade-off, the resulting generalization bounds are also in expectation over both data and models, which is a weaker notion of generalization. In contrast to the Bregman Misfit-Gain Inequality in Mulgund & Pabbaraju (2025) where the "residual" term $\epsilon$ has a clear interpretation as the estimation+optimization error, the residual terms in Theorem 3.1, $\epsilon_1, \epsilon_2$, defined based on expectations over both data and models, and therefore the entire training process, are less interpretable. This trade-off is not sufficiently discussed in the paper.
    - Yao et al., 2025 has provided the theoretical and empirical study for reversed KL/CE in W2SG. Section 5 echoes that story, but the novelty of this work along this direction seems limited.
    - Overall, conditioned on the context provided by the prior works, the technical and intuition-level contributions of this paper seem incremental and a combination of prior results up to small variations.
3. The relaxation of the convexity assumption is the key technical contribution of this work. Regardless of whether this relaxation under the trade-off discussed above is significant enough, the benefit of such relaxation is not well illustrated rigorously, except for the obvious argument that non-convex students are common in practice. In particular, the only tractable theorem, Theorem 4.1, assumes random features + ridge head, which is still convex and essentially linear.
4. On Theorem 4.1 itself, while it is common and reasonable to assume Gaussian data/features in random matrix theory-based analyses, assuming that the (potentially pre-trained) model weights are also Gaussian is very strong and unrealistic. This seems to be another artifact caused by the joint expectation over data and models.

I am happy to rectify any misunderstandings of the paper and adjust my assessment if the authors can provide clarifications and counter-arguments for the above points.

**Questions:**

Below are some minor questions and comments noted during my reading:
1. Lines 51-53: "... the misfit error simultaneously quantifies the attainable performance gain, implying that zero misfit error is in fact undesirable; the work does not explicitly discuss in what sense the student should align with the teacher." The sentence is quite confusing in the context: what "the misfit error simultaneously quantifies the attainable performance gain" means, simultaneously with respect to what, what is an implicit vs. explicite description of the alignment between student and teacher?
2. Lines 69-70: "... where the posterior is defined by conditioning the teacher’s distribution on the student model, and the mean may involve a dual expectation." This sentence is a bit hard to parse; what is meant by "conditioning the teacher’s distribution on the student model," and what does "dual expectation" mean here?
3. Lines 73-74: "... we show that reducing the expected misfit between teacher and student is sufficient for W2SG." This sentence is somehow confusing. Given the context of Charikar et al. (2024) and Mulgund & Pabbaraju (2025), I guess "reducing the expected misfit between teacher and student" refers to the process of aligning the student to the teacher? Notice that the misfit-gain inequalities, e.g., in Mulgund & Pabbaraju (2025) and your Theorem 3.1, also have an expected misfit term, reducing of which actually improves the upper bound on the student risk. It would be good to clarify the meaning of "misfit" in the context.
4. The definition of dual expectation in Lemma 2.1 is not rigorous or clear in the context.

---

> ### Author Response · Authors · 2025-11-21
> **Response to Reviewer GSgt (Part 1/5)**
>
> We thank the reviewer for the valuable feedback and greatly appreciate your openness to re-evaluate our paper based on our rebuttal. Our responses follow.
>
> >W1: While the detailed discussion on the relation with prior works and the technical novelties upfront in the introduction (lines 46-82) can provide good motivations for this work for domain experts (as mentioned in Strengths), it could be overwhelming for general audience in the community. I would suggest reorganizing the introduction carefully, e.g., partitioning the technical novelties in lines 60-82 into bullet points, each summarized by a concise, intuitive title explaining the effect/benefit.
>
> **Response:** Thanks for your suggestion. We recognize that the current writing may be overwhelming for general audience.
>
> We have revised the introduction (Lines 83–101) according to your suggestion. Here, we summarize the structure and main insights of the paper:
>
> 1. We establish W2SG inequalities without convexity assumptions based on a generalized bias–variance decomposition of Bregman divergence, and further derive a sufficient condition for W2SG to occur—namely, that the student model aligns with its posterior mean teacher. Under symmetric Bregman divergence (i.e., squared loss), this implies that a sufficiently large student model ensures W2SG.
> 2. Extending to asymmetric Bregman divergence, we show that increasing the student’s predictive confidence (reducing entropy) promotes W2SG. Furthermore, we theoretically demonstrate that RCE is robust to the teacher's predictive uncertainty.
> 3. We empirically verify the theoretical sufficient conditions for W2SG and the robustness of RCE, and propose CACE loss to effectively leverage information from the teacher's label confidence.
>
> We thank the reviewer again for the helpful suggestion, which has significantly improved the clarity of our introduction.
>
>
> >W2: (1) From a high-level perspective, while I acknowledge the technical novelties in the main theoretical results, I feel that the significance of the improvements brought by these novelties is not sufficient to meet the bar of ICLR. In particular:
> >Mulgund & Pabbaraju, 2025 already extend misfit-gain to general Bregman losses and propose convex combinations of logistic heads for classification; this paper avoids convexity by going to the expected risk over both data and models. While the main results in Theorem 3.1 relaxes the key limitation of the convexity assumption in prior works, as a trade-off, the resulting generalization bounds are also in expectation over both data and models, which is a weaker notion of generalization.
>
> **Response:** We sincerely thank the reviewer for acknowledging our technical novelties.
>
> However, we respectfully disagree with the characterization of our formulation as a "weaker notion of generalization". On the contrary, we view it as a strength, as it explicitly captures the statistical dependency between the teacher and student models (i.e. the dependence between the random varaibles $W$ and $W'$), which previous works overlook. Importantly, please note that we follow the standard bias-variance decomposition for the population risk [1], which is essentially the expected version of risk over both data and models, naturally accounting for the randomness of training set selection and learning algorithm, and thus aligns with the standard settings in ML theory.
>
> Moreover, this formulation avoids the necessity of imposing strong geometric constraints on the hypothesis class. Specifically, in Mulgund & Pabbaraju (2025), the student model is not an arbitrary choice, but is required to coincide with or be close to the projection of the teacher onto a convex space.
>
> Finally, note that our expected W2SG inequality can be converted into a high-probability bound via standard concentration arguments, which is the usual way to translate an expectation-level statement into a high-probability guarantee.
>
> [1] Bishop, C. M. (2006). Pattern Recognition and Machine Learning.

---

> ### Author Response · Authors · 2025-11-21
> **Response to Reviewer GSgt (Part 2/5)**
>
> >W2: (2) In contrast to the Bregman Misfit-Gain Inequality in Mulgund & Pabbaraju (2025) where the "residual" term $\epsilon$ has a clear interpretation as the estimation+optimization error, the residual terms in Theorem 3.1, $\epsilon_1$ ,$\epsilon_2$, defined based on expectations over both data and models, and therefore the entire training process, are less interpretable.
>
> **Response:** We note that the residual term in Mulgund & Pabbaraju (2025) involves an implicit dependence on an $\epsilon$–$\delta$ trade-off, and the condition for their residual term to vanish is highly restrictive: the student model must coincide with the convex projection of the teacher onto the student's function class. Furthermore, we are unclear why the reviewer regards their $\epsilon$ as the "estimation + optimization error," as the original paper does not explicitly describe it as such.
>
> In contrast, our Thm. 3.1 provides a closed-form characterization of the residual terms $\epsilon_1$ and $\epsilon_2$. We further analyze these terms in Cor. 4.1 and Thm. 4.1, which give explicit sufficient conditions under which $\epsilon_2$ (and $\epsilon_1$) vanish, namely, when the student model is sufficiently large, ensuring convergence toward its posterior mean teacher. **In this sense, our results are more interpretable and informative than prior work**: previous misfit-based analyses do not quantitatively demonstrate that increasing student model size can reduce the residual term, a conclusion that critically relies on having a closed-form expression such as the one provided in our Thm. 3.1.
>
> >W2: (3) This trade-off is not sufficiently discussed in the paper.
>
> **Response:** As also mentioned above, we do not view the use of expectation over both data and models as a trade-off. Instead, we consider it a fundamental difference from prior works, as noted in Lines 66–68. To make this distinction more clear, we have revised the paper (Lines 218–221) to explicitly highlight this point in the explanation of Thm. 3.1.
>
>
> >W2: (4) Yao et al., 2025 has provided the theoretical and empirical study for reversed KL/CE in W2SG. Section 5 echoes that story, but the novelty of this work along this direction seems limited.
>
> **Response:** While Yao et al. (2025) primarily focus on the mode-seeking property of RKL and mitigating the influence of unreliable weak supervision, our work investigates W2SG from a completely different perspective, providing clear theoretical and empirical innovations.
>
> Theoretically, from an optimization perspective we show that RCE is significantly less sensitive to the predictive uncertainty of the teacher (Prop. 2) and can achieve larger performance gains compared to CE (Prop. 1). Moreover, Yao et al. (2025) still rely on the strong convexity assumption inherited from prior misfit-based analyses, whereas our Cor. 5.1 overcomes this and additionally provides the new insight that high-confidence predictions by the student are beneficial for outperforming the teacher.
>
> Empirically, we not only validate RCE's robustness to low-confidence labels, but also leverage this property to propose CACE, demonstrating its superior performance across multiple W2SG tasks.
>
> All of these contributions are not covered in Yao et al. (2025), and therefore we do not consider the novelty of our work along this direction to be limited.

---

> ### Author Response · Authors · 2025-11-21
> **Response to Reviewer GSgt (Part 3/5)**
>
> >W2: (5) Overall, conditioned on the context provided by the prior works, the technical and intuition-level contributions of this paper seem incremental and a combination of prior results up to small variations.
>
> **Response:** We respectfully disagree with the assessment that our technical and intuition-level contributions are merely incremental or simple combinations of prior results up to small variations.
>
> We also note that this comment appears inconsistent with the reviewer’s earlier acknowledgment of the "technical novelties in the main theoretical results." To clarify, our technical developments are distinct from prior misfit-based analyses. Unlike Mulgund & Pabbaraju (2025), which rely on the generalized Pythagorean theorem, our approach is built upon a generalized bias–variance decomposition of Bregman divergences, allowing us to completely remove the convexity, realizability, and sequential consistency assumptions in prior works. In addition, by leveraging random matrix theory–based analyses of double descent in Thm. 4.1, we obtain new insights on how the student model size influences W2SG—an aspect not explored in prior studies.
>
> Beyond the technical developments, our intuition-level insights are genuinely new and, to the best of our knowledge, absent from previous misfit-based results. For example, Thm. 4.1 shows that increasing the student model size effectively drives the student toward the posterior mean teacher, providing a clear explanation of when W2SG is guaranteed to occur. Similarly, Cor. 5.1 demonstrates that reducing the entropy of the student's prediction is beneficial for outperforming the teacher.
>
> None of these insights are stated or implied from misfit-based prior works, and together they provide a coherent and novel conceptual understanding of W2SG rather than small variations of existing results.
>
> >W3: The relaxation of the convexity assumption is the key technical contribution of this work. Regardless of whether this relaxation under the trade-off discussed above is significant enough, the benefit of such relaxation is not well illustrated rigorously, except for the obvious argument that non-convex students are common in practice. In particular, the only tractable theorem, Theorem 4.1, assumes random features + ridge head, which is still convex and essentially linear.
>
> **Response:** We first appreciate the reviewer’s recognition of our contribution in relaxing the convexity assumption.
>
> Indeed, the reviewer raises a valid point, and we acknowledge that the example in Thm. 4.1 operates in a convex setting. However, as existing double-descent theory is still mostly restricted to random-feature models and other convex hypothesis classes, the theoretical tools needed for non-convex cases are not yet fully developed, establishing an analogous result for non-convex hypothesis classes is highly challenging.
>
> If double-descent theory were to achieve breakthroughs in the non-convex setting, our framework would naturally yield insights far more general than prior misfit-based analyses and could also contribute to advancing the broader theoretical understanding of double descent.
>
> In addition, Thm. 4.1 is not inherently limited to linearity. It can be extended to two-layer random-feature models by leveraging asymptotic analysis techniques common in the overparameterized regime. This extension is supported by [1], whose findings indicate that enlarging the model size in overparameterized random-feature networks continues to reduce test error—consistent with the conditions under which W2SG arises.
>
> [1] Mei et al., "The Generalization Error of Random Features Regression: Precise Asymptotics and the Double Descent Curve." In Communications on Pure and Applied Mathematics 2022.

---

> ### Author Response · Authors · 2025-11-21
> **Response to Reviewer GSgt (Part 4/5)**
>
> >W4: On Theorem 4.1 itself, while it is common and reasonable to assume Gaussian data/features in random matrix theory-based analyses, assuming that the (potentially pre-trained) model weights are also Gaussian is very strong and unrealistic. This seems to be another artifact caused by the joint expectation over data and models.
>
> **Response:** We acknowledge that the Gaussian assumption on the model is a limitation. However, such limitation is not specific to our analysis, nor is it an artifact of taking expectations over both data and models. Rather, it is a common constraint shared by nearly all existing theoretical works on W2SG, including [1,2,3], where either the features or model weights are assumed to follow Gaussian distributions.
>
> These studies adopt such assumptions primarily to ensure theoretical tractability. Our Thm. 4.1 aligns with these works by making similar assumptions to simplify the analysis, enabling us to derive clean asymptotic solutions that provide clear insight into the effect of model capacity.
>
> This is a general limitation of current theoretical approaches to W2SG, and we have clarified this in the revised conclusion section (Lines 496–499).
>
> [1] Somerstep et al., "A Transfer Learning Framework for Weak-to-strong Generalization." In ICLR 2025.
>
> [2] Wu and Sahai, "Provable Weak-to-Strong Generalization via Benign Overfitting." In ICLR 2025.
>
> [3] Dong et al., "Discrepancies are Virtue: Weak-to-Strong Generalization through Lens of Intrinsic Dimension." In ICML 2025.
>
> >Q1: Lines 51-53: "... the misfit error simultaneously quantifies the attainable performance gain, implying that zero misfit error is in fact undesirable; the work does not explicitly discuss in what sense the student should align with the teacher." The sentence is quite confusing in the context: what "the misfit error simultaneously quantifies the attainable performance gain" means, simultaneously with respect to what, what is an implicit vs. explicit description of the alignment between student and teacher?
>
> **Response:** We appreciate the opportunity to clarify this sentence. The statement that “the misfit error simultaneously quantifies the attainable performance gain” means that, under Cor. 3.1, when the remainder term $\epsilon$ vanishes, the entire performance gap between the student and the teacher is exactly given by the misfit term. In this sense, the magnitude of the misfit directly quantifies the attainable performance gain.
>
> The "simultaneously" describes the dual role of the misfit in W2SG. On one hand, minimizing misfit facilitates the emergence of W2SG, as Eq. (8) shows that reducing misfit is a sufficient condition to reduce the residual $\epsilon$ term. On the other hand, since the misfit term effectively upper-bounds the performance gain, driving it completely to zero is undesirable as it would imply zero improvement over the teacher.
>
> Furthermore, regarding "explicit description of the alignment between student and teacher," our Thm. 4.1 provides a more explicit characterization: increasing the student model size reduces the $\epsilon$ term, making W2SG more likely to occur. This offers a more explicit and interpretable alignment condition than Mulgund & Pabbaraju (2025), which requires the student to approach the convex projection of the teacher within the student class.

---

> ### Author Response · Authors · 2025-11-21
> **Response to Reviewer GSgt (Part 5/5)**
>
> >Q2: Lines 69-70: "... where the posterior is defined by conditioning the teacher’s distribution on the student model, and the mean may involve a dual expectation." This sentence is a bit hard to parse; what is meant by "conditioning the teacher’s distribution on the student model," and what does "dual expectation" mean here?
> >
> >Q4: The definition of dual expectation in Lemma 2.1 is not rigorous or clear in the context.
>
> **Response:** We thank the reviewer for pointing this out. To clarify, Lines 69–70 are intended to summarize the core insight of Cor. 3.1: W2SG is guaranteed to emerge when the student aligns with its "posterior mean" teacher $\mathbb{E}[f _ {W}(x)\mid W']$ or its dual “posterior mean” teacher $\mathcal{E}[f _ {W}(x)\mid W']$. Notably, under symmetric Bregman divergence (i.e., squared loss) these two quantities are identical.
>
> The phrase "conditioning the teacher’s distribution on the student model" refers to the conditional distribution $P _ {W\mid W'}$. As discussed in Remark 2.1, the teacher $W$ and student $W'$ are inherently dependent because the student is trained on the teacher’s pseudo-labels. This dependency induces a joint distribution $P _ {W,W'}$, making the conditional distribution $P _ {W\mid W'}$—which we refer to as the "posterior"—well-defined.
>
> We formally define the dual expectation in Lemma 2.1 as $\mathcal{E}[X] \triangleq (\mathbb{E}[X^\*])^\*$. Here, $x^\* = \nabla \phi(x)$ represents the dual coordinate. By the properties of the convex conjugate, the inverse mapping is given by $x = (x^\*)^\*=\nabla\phi^\*(\nabla\phi(x))$. Consequently, the definition is equivalent to:
>
> $$
> \mathcal{E}[X] \triangleq \nabla \phi^*(\mathbb{E}[\nabla \phi(X)]) \quad \Longleftrightarrow \quad \nabla \phi(\mathcal{E}[X]) = \mathbb{E}[\nabla \phi(X)]
> $$
>
> This quantity is closely related to the minimizers of the expected Bregman divergence. Specifically, as shown in Lemma 0.1 of Pfau (2013), the minimizer of the "right-sided" expected Bregman divergence satisfies
>
> $$
> z = \arg\min _ {y\in\mathbb{R}^d} \mathbb{E}[D _ {\phi}(y,X)] \Leftrightarrow \nabla \phi(z) = \mathbb{E}[\nabla \phi(X)]
> $$
>
> Thus, the dual expectation can be equivalently expressed as
>
> $$
> \mathcal{E}[X] = \arg\min _ {y\in\mathbb{R}^d} \mathbb{E}[D _ {\phi}(y, X)]
> $$
>
> In contrast, the standard expectation $\mathbb{E}[X]$ minimizes the "left-sided" form $\mathbb{E}[D _ \phi(X, y)]$. We have added Definition 2.2 (Lines 163–165) to more explicitly define the dual expectation.
>
>
> >Q3: Lines 73-74: "... we show that reducing the expected misfit between teacher and student is sufficient for W2SG." This sentence is somehow confusing. Given the context of Charikar et al. (2024) and Mulgund & Pabbaraju (2025), I guess "reducing the expected misfit between teacher and student" refers to the process of aligning the student to the teacher? Notice that the misfit-gain inequalities, e.g., in Mulgund & Pabbaraju (2025) and your Theorem 3.1, also have an expected misfit term, reducing of which actually improves the upper bound on the student risk. It would be good to clarify the meaning of "misfit" in the context.
>
> **Response:** We thank the reviewer for the helpful question and clarify as follows.
>
> The misfit term involves a subtle trade-off. On one hand, as shown in Section 4, reducing the expected misfit decreases the residual term $\epsilon$—specifically, when the misfit approaches the conditional variance in Eq. (8), the $\epsilon$ term vanishes entirely. On the other hand, since the misfit also upper-bounds the attainable performance gain, reducing it excessively could theoretically reduce the magnitude of improvement.
>
> However, our analysis focuses on the sufficient condition for W2SG to emerge. In this context, reducing the misfit (towards the conditional variance) is sufficient to drive the residual $\epsilon$ to zero, thereby guaranteeing that W2SG occurs (i.e., a non-zero gain exists).
>
> We agree with the reviewer’s intuition that "reducing the expected misfit ... refers to the process of aligning the student to the teacher." The misfit term utilized in this optimization process conveys the same fundamental concept as the misfit term in the misfit-gain inequalities, differing only in that one is empirical and the other is population risk.
>
> To prevent confusion, we have revised the manuscript (Lines 71–73) to make this distinction explicit.

---

> > ### Comment · Reviewer_GSgt · 2025-11-27
> >
> > Thank you for the careful and detailed responses. It partially addressed my questions.
> >
> > However, my main concern remains the incremental nature of the contribution relative to the existing W2SG and misfit-based literature. To clarify, when I said in my original review that I “acknowledge the technical novelties,” I meant that I understand how results in this work differ from works such as Charikar et al., Mulgund & Pabbaraju, and Yao et al., not that I regard these differences as particularly informative or sufficiently significant for a venue like ICLR. I think the move to expectation over both data and models to remove the convexity assumptions, as well as the random-feature/double-descent instantiation, provides a refinement of and variation on existing theory rather than bringing substantially new perspectives or qualitatively stronger guarantees. The amount of effort required to articulate and position these distinctions against prior work reinforces my impression that the overall contribution is incremental.
> >
> > I therefore still believe the work falls below the bar for acceptance at ICLR. I understand that this evaluation can be subjective, and I have no intention of continuing to iterate on this point. I will raise my rating to 4, for the improved presentation only.

---

> > > ### Author Response · Authors · 2025-11-28
> > > **Response to Reviewer GSgt**
> > >
> > > We would like to thank the reviewer for reading our rebuttal and for raising the score in recognition of the improved presentation. While we fully respect the reviewer's choice not to engage in further discussion, we feel it is still important to provide additional clarification so that the AC and other reviewers can clearly understand the points of disagreement.
> > >
> > >
> > > First, we must firmly and respectfully disagree with the claim of our work as merely incremental, as this overlooks the core technical advances we provide. Notably, the previous misfit-based results can **be directly recovered** from our main theorem (see Remark A.1 in Appendix A) under the same assumptions, demonstrating that our misfit-based framework is strictly more general than previous works. Moreover, we believe that describing the removal of convexity assumptions as only "a refinement of and variation on existing theory" is somewhat biased. The non-convex nature of DNNs is well known, and there is no reason for insisting on convexity-based arguments. While we acknowledge that our work does not yet give a complete story of practical W2SG, removing this unrealistic assumption is undeniably a meaningful and sustainable advancement in misfit-based explanations of W2SG.
> > >
> > >
> > > Second, regarding the statement that our differences from prior works are "not particularly informative or sufficiently significant", since the reviewer does not provide concrete reasons for this assessment, for the benefit of the AC and remaining reviewers, we reiterate that the insights provided by our work, **none of which appear in prior literature**. In particular, we identify that W2SG emerges when the student approximates the posterior mean teacher, and that increasing the student model size encourages convergence toward this posterior mean. This gives a concrete and interpretable statistical explanation that goes well beyond the previous notion of "projection onto a convex set". Moreover, we develop theoretical analysis within our misfit framework regarding student predictive confidence, and we present novel findings on RCE robustness and algorithmic improvements. These contributions represent new perspectives and explanatory mechanisms, rather than refinements of existing results.
> > >
> > > Regarding the technical novelty, we appreciate that the reviewer acknowledges the novel development of our theoretical results, and it appears there is no disagreement on this point.
> > >
> > > We hope that our response clarifies the non-triviality and significance of our contributions, and is helpful for the AC–reviewer discussion. We still want to thank Reviewer GSgt for many valuable suggestions that improved the presentation of our work. Should the reviewer reconsider and wish to continue the discussion, we remain fully willing to provide any further clarification.

---

### Official Review · Reviewer_MAkU · 2025-11-01

**Soundness:** 3
**Presentation:** 2
**Contribution:** 3
**Rating:** 4
**Confidence:** 4

**Summary:**

The paper considers an abstract weak-to-strong learning setup and shows a misfit-based inequality characterizing the W2S improvement in expected population risks, which removes the convexity assumption from the previous misfit-based W2S inequalities. From this inequality, they show how aligning the student with the posterior mean teacher can improve W2S performance gain. They also apply their inequality to two settings: (i) an overparametrized ridge regression setting in the appropriate scaling limit of number of weak samples vs the teacher and student dimensions, which indicates that W2S improvement is likely to emerge with a student of higher capacity, and (ii) to the case when risk is measured using CE or RCE, showing that RCE is better as optimizing objective. Finally, they use these insights to empirically show improvements in some cases W2S when combining the two.

**Strengths:**

The misfit based approach to W2S generalization is a promising approach that can naturally also model pre-training and encompasses a lot of different model classes. The generalization to non-convex student hypothesis class is a significant improvement which comes from the idea of analyzing expected population risk using the bias-variance decomposition of Bergman divergences, which is an original idea.


The mathematical statements are clear and on the first reading seem correct.


The main significance of results is that:
1. it removes one of the strongest assumptions from previous misfit-based approaches
2. provides some insights into the W2S mechanism through the example of ridge regression, suggesting that higher capacity students might be beneficial
4. theoretically suggest the benefits of using RCE as optimization objective, demonstrating why some form of regularization on the student is necessary for W2S generalization
5. empirically improves performance of W2S generalization in some setups by combining CE and RCE.

**Weaknesses:**

1. One of the main messages of the paper, that W2S is more likely to occur by mimicking the posterior teacher or the algorithmic encouragement to train on an ensemble of teachers are not in the spirit of W2S generalization and have the effect of confounding other effects with W2S. Consider regression setup: of course that training on a less noisy distribution is beneficial. Averaging over many teachers effectively denoises the teacher predictor artificially which makes the student have smaller error. If you take very large number of teachers, even training the student suboptimally (i.e. to harmfully overfit) without regularization (which would make the student effectively mimic the teacher since it can fit the teacher predictor) with an ensemble of teachers would make the student outperform one teacher because of the denoising effect. However, the question of W2S-like generalization with multiple teachers is still interesting in its own right, but for the sake of understanding the mechanisms in W2S it is important to not confuse the two.

2. The benefit of removing the convexity assumption on the student class is not demonstrated. The new misfit based inequality, whose main novelty is circumventing the convexity assumptions on the student, i.e. Theorem 3.1 and Corollary 3.1, give sufficient conditions for W2S but do not establish that W2S actually occurs. The paper only establishes that these inequalities do imply that W2S generalization occurs in the setting of ridge regression in Theorem 4.1, but (i) in this case the convexity assumption on the loss is satisfied and (ii) it was already known that W2S generalization can happen in this case (e.g. Wu & Sahai 2025). In the CE or RCE case the W2S generalization is not actually established, only the sufficient conditions in Corollary 5.1. So the benefit of removing the convexity assumption is unclear.

3. Some of the main contribution claims are not precise and should be further discussed. E.g: the claim that “W2S is more likely to emerge when the student has higher capacity” is a bit unclear and goes against other conclusions in the paper that not overfitting to the teacher is important. It is also not always the case in practice, e.g. in the original Burns et al 2023 paper, the authors show that on some tasks the smaller student models are better for W2S (Figure 3, chess example).

**Questions:**

1. What are the main drawbacks of using an ensemble of teachers? Can we use the tools developed in the paper to theoretically analyze the case with an ensemble of teachers?

2. Are there any simple examples with a student with a non-convex hypothesis class where Theorem 3.1 can help establish W2S?

3. In regression example in Theorem 4.1, is the teacher predictor optimal (for its set of features/kernel)? Does the W2S improvement come from the teacher being suboptimally regularized and the student being optimally regularized or no? If not, where does it come from?

---

> ### Author Response · Authors · 2025-11-21
> **Response to Reviewer MAkU (Part 1/3)**
>
> We thank you sincerely for your valuable feedback on our paper. Our responses follow.
>
> >W1: One of the main messages of the paper, that W2S is more likely to occur by mimicking the posterior teacher or the algorithmic encouragement to train on an ensemble of teachers are not in the spirit of W2S generalization and have the effect of confounding other effects with W2S. Consider regression setup: of course that training on a less noisy distribution is beneficial. Averaging over many teachers effectively denoises the teacher predictor artificially which makes the student have smaller error. If you take very large number of teachers, even training the student suboptimally (i.e. to harmfully overfit) without regularization (which would make the student effectively mimic the teacher since it can fit the teacher predictor) with an ensemble of teachers would make the student outperform one teacher because of the denoising effect. However, the question of W2S-like generalization with multiple teachers is still interesting in its own right, but for the sake of understanding the mechanisms in W2S it is important to not confuse the two.
>
> **Response.** We thank the reviewer for the careful reading, yet we would first like to clarify that our work focuses on identifying sufficient conditions under which W2SG occurs. In particular, Section 3 makes no assumptions beyond the basic setup, and Thm. 3.1 together with Cor. 3.1 shows that the student approaching the posterior mean teacher is one such sufficient condition.
>
> After Section 3, our Thm. 4.1 further characterizes how a student can approach its posterior mean teacher, demonstrating that increasing the student model size is sufficient. Therefore, the use of an ensemble of teachers is not a central message of our main theorem, what we truly aim to highlight is that increasing the student model size enables it to approach its posterior mean teacher.
>
> That said, we do empirically verify the benefit of using teacher ensembles (as a practical proxy for the posterior mean teacher), which follows directly as an implication of our theory.
>
> Moreover, we do not claim that using an ensemble of teachers would "make the student outperform one teacher"; rather, please note that the improvement we refer to is in terms of expected performance, not dominance over a specific teacher.
>
> Nevertheless, we agree that the reviewer's intuition on using model ensembles to denoise the teacher predictor, and more generally on W2S-like generalization with multiple teachers, is interesting. In our experiments, approximating the posterior teacher with ensembles improves W2SG performance primarily by reducing variance, consistent with this intuition.
>
> >Q1: What are the main drawbacks of using an ensemble of teachers? Can we use the tools developed in the paper to theoretically analyze the case with an ensemble of teachers?
>
> **Response.** The main drawback of using an ensemble of teachers is the increased computational cost, as it requires training multiple weak teacher models.
>
> Furthermore, the tools developed in the paper can be used to theoretically analyze the effect of ensemble supervision. Here, we theoretically interpret the reviewer's "ensemble of teachers" as the expected teacher model $\mathbb{E}_W[f_W(x)]$, which differs from the posterior mean teacher $\mathbb{E}_W[f_W(x)\mid W']$ in Thm. 3.1. Notably, we address this expected teacher model explicitly in Thm. B.1 (See Appendix B.2), which shows that the student should be sufficiently close to the expected teacher (empirically approximated via a model ensemble) to ensure a performance improvement.  Additionally, from an information-theoretic perspective, the gap between the misfit terms in Thm. 3.1 and Thm. B.1 can be characterized by the mutual information between strong and weak models (we can elaborate on this upon request).
>
> If our interpretation is not fully aligned with the reviewer's intent, we are happy to discuss this further.

---

> ### Author Response · Authors · 2025-11-21
> **Response to Reviewer MAkU (Part 2/3)**
>
> >W2: The benefit of removing the convexity assumption on the student class is not demonstrated. The new misfit based inequality, whose main novelty is circumventing the convexity assumptions on the student, i.e. Theorem 3.1 and Corollary 3.1, give sufficient conditions for W2S but do not establish that W2S actually occurs. The paper only establishes that these inequalities do imply that W2S generalization occurs in the setting of ridge regression in Theorem 4.1, but (i) in this case the convexity assumption on the loss is satisfied and (ii) it was already known that W2S generalization can happen in this case (e.g. Wu & Sahai 2025). In the CE or RCE case the W2S generalization is not actually established, only the sufficient conditions in Corollary 5.1. So the benefit of removing the convexity assumption is unclear.
> >
> >Q2: Are there any simple examples with a student with a non-convex hypothesis class where Theorem 3.1 can help establish W2S?
>
> **Response:** We thank the reviewer for raising this important point.
>
> We acknowledge that the example in Thm. 4.1 operates in a convex setting. However, we respectfully disagree that our conclusions are covered by Wu & Sahai (2025).  While both works analyze linear models, the theoretical mechanisms and assumptions differ substantially. Our analysis identifies student model capacity as a sufficient driver for W2SG: we prove that simply increasing the student's over-parameterization ratio leads to convergence toward the teacher's posterior mean, thereby guaranteeing W2SG. In contrast, Wu & Sahai (2025) primarily attribute the phenomenon to phase transitions in the amount of weakly labeled data and do not emphasize the role of model size. Moreover, the two works rely on different assumptions concerning the data distribution and the definition of W2S model relations.
>
> Second, although for CE and RCE we do not yet provide a non-convex analogue of Thm. 4.1, our analysis still yields concrete and practically relevant insights, most notably, that high-confidence student predictions facilitate outperforming the teacher (Cor. 5.1). These insights are not captured by prior misfit-based analyses.
>
> Regarding the convexity setup, our reliance on the linear setting in Thm. 4.1 reflects the current limitations of existing double descent theory, which largely focuses on random-feature or other convex-class models. Extending these results to non-convex settings, especially under cross-entropy loss, remains an open challenge, despite extensive empirical evidence of double descent in the classification task. Thus, establishing a result analogous to Thm. 4.1 beyond convex classes would likely require substantial advances in our theoretical understanding of double descent under these losses.
>
> We hope the reviewer understands that deriving such results is inherently difficult given the present state of DL theory. Nevertheless, using Thm. 4.1 to illustrate the insight given by a misfit-based W2S inequality is already a novel contribution: previous misfit-based analyses have not provided an explicit example demonstrating how model size influences the emergence of W2SG. Importantly, as theoretical tools for nonconvex cases continue to advance, Thm. 4.1 can be readily extended.
>
> At the same time, our contributions go well beyond Thm. 4.1, providing new sufficient conditions and conceptual insights that were not available in previous misfit-based work.

---

> ### Author Response · Authors · 2025-11-21
> **Response to Reviewer MAkU (Part 3/3)**
>
> >W3: Some of the main contribution claims are not precise and should be further discussed. E.g: the claim that “W2S is more likely to emerge when the student has higher capacity” is a bit unclear and goes against other conclusions in the paper that not overfitting to the teacher is important. It is also not always the case in practice, e.g. in the original Burns et al 2023 paper, the authors show that on some tasks the smaller student models are better for W2S (Figure 3, chess example).
>
> **Response.** We give the following clarification regarding the relationship between student capacity and overfitting, as these two aspects play distinct but complementary roles in W2SG:
>
> From a model-capacity perspective: Increasing the student size strengthens its ability to fit the posterior mean teacher, thus serving as a sufficient condition for W2SG to occur (i.e., achieving a non-zero performance gain). From a practical training perspective, preventing the student from perfectly mimicking the imperfect teacher (i.e., avoiding overfitting) ensures that the misfit term remains large and positive, which is the source of the performance gain.
>
> Thus, student size and overfitting are not contradictory factors: the former facilitates the emergence of W2SG, while the latter influences the magnitude of the resulting improvement. In other words, if we only care about whether W2SG occurs, simply increasing the student’s size is sufficient. However, if we hope for the student to achieve optimal performance under weak-teacher supervision, we should avoid having the student overfit completely to the teacher. In fact, we do admit that there often exists an optimal student size that maximizes W2S performance, which is consistent with the empirical observations reported in Burns et al. (2023).
>
> We have revised Remark 4.1 (Line 309-310) in the paper to avoid potential misunderstandings.
>
>
> >Q3: In regression example in Theorem 4.1, is the teacher predictor optimal (for its set of features/kernel)? Does the W2S improvement come from the teacher being suboptimally regularized and the student being optimally regularized or no? If not, where does it come from?
>
> **Response.** First, in our construction, the emergence of W2SG depends solely on the intrinsic relationship between the student and the posterior mean teacher, and is independent of whether the teacher predictor is optimal, as we impose no constraints on the ground truth labeling function $g$.  This distinguishes our work from Wu & Sahai (2025) (indeed,  in Example 1, no assumption on $g$ is required, which we view as a strength of our analysis).
>
> Specifically, this improvement arises when the student becomes sufficiently close to the posterior mean teacher. By Eq. (8), their distance decomposes into the expected misfit minus the conditional variance. Furthermore, Thm. 4.1 and Cor. 4.2 show that with sufficiently large student capacity, the misfit term approaches the conditional variance and remains positive, thereby guaranteeing the emergence of W2SG.
>
> This reveals the core insight of our analysis: the conditions under which W2SG appears are governed by the generalization behavior of highly overparameterized students, directly connecting our results to the double descent theory.
>
> In summary, Thm. 4.1 demonstrates that the W2S improvement originates from student capacity, rather than from teacher or student regularization.

---

> ### Author Response · Authors · 2025-11-28
>
> Dear Reviewer MAkU,
>
> Thank you for your time and effort in reviewing our paper. As the discussion period is approaching its end, we wish to confirm whether our responses have adequately addressed your concerns. We would greatly appreciate it if you could re-evaluate our paper based on our rebuttal.
>
> Thank you again for your consideration and support.
>
> Best regards, Paper 15511 Authors

---

### Official Review · Reviewer_xcki · 2025-11-04

**Soundness:** 3
**Presentation:** 3
**Contribution:** 3
**Rating:** 6
**Confidence:** 2

**Summary:**

This paper provides a theoretical analysis of weak-to-strong generalization (W2SG)—the phenomenon where a strong student model trained on weak teacher labels surpasses the teacher's performance. Using a generalized bias-variance decomposition of Bregman divergence, the authors prove that the performance gap is quantified by the expected misfit between models, removing restrictive convexity assumptions from prior work. Key insights include: students should approximate the posterior mean teacher rather than individual teachers, reverse cross-entropy is less sensitive to teacher uncertainty, and avoiding overfitting facilitates W2SG. Empirical results validate the theoretical findings.

**Strengths:**

- The paper studies an interesting problem and offer a theoretically driven explanation to the weak-to-strong generalization phenomenon observed.
- The authors of the paper propose a novel reverse cross-entropy loss and empirically demonstrate the effectiveness of it.

**Weaknesses:**

- The experiments conducted use relatively small models, by today's standard. While I understand the computational resource constraint, it would be interesting to use more recent and larger models.
- To the best of my understanding, the analysis is done with respect to linear models.

**Questions:**

- Is there a particular reason why different models are used with different methods independently in Table 1? Do the conclusions hold beyond the combinations shown?

---

> ### Author Response · Authors · 2025-11-21
> **Response to Reviewer xcki**
>
> We thank you sincerely for your valuable feedback on our paper. Our responses follow.
>
> > W1: The experiments conducted use relatively small models, by today's standard. While I understand the computational resource constraint, it would be interesting to use more recent and larger models.
> >
> > Q1: Is there a particular reason why different models are used with different methods independently in Table 1? Do the conclusions hold beyond the combinations shown?
>
> **Response:** Thank you for the suggestions regarding the experiments.
>
> First, our validation experiments and algorithm comparisons already include models up to 14B parameters. To the best of our knowledge, virtually no prior W2SG theoretical work has been empirically evaluated at this scale. Further increasing the model size is challenging due to computational resource constraints, and we hope the reviewer can understand this point.
>
> In addition, our conclusions hold beyond the specific combinations shown. We have supplemented our paper with new experiments on the CAI-Harmless and HH-RLHF datasets using the Qwen series, shown in Table 1. The results further confirm the advantages of CACE and provide stronger empirical evidence supporting our conclusions.
>
>
> >W2: To the best of my understanding, the analysis is done with respect to linear models.
>
> **Response.** While Thm. 4.1 uses ridge regression in a linear model to illustrate how the model size affects the emergence of W2SG, it is generalizable to non-linear architectures. As noted in Lines 311–312, Thm. 4.1 can be extended to non-linear neural networks, specifically two-layer random-feature networks, by employing asymptotic analysis techniques common in the overparameterized regime. This extension is supported by the double-descent analyses of [1], whose results show that in overparameterized random-feature models, increasing the model size continues to reduce the test error—consistent with the conditions under which W2SG arises. Furthermore, extending the theoretical scope to deeper non-linear networks is feasible within the Neural Tangent Kernel (NTK) regime [2,3,4]. In summary, although presented via a linear model for analytic clarity, the conclusion that sufficient model capacity promotes W2SG continues to hold across a broad range of non-linear network architectures.
>
> [1] Mei et al., "The Generalization Error of Random Features Regression: Precise Asymptotics and the Double Descent Curve." In Communications on Pure and Applied Mathematics 2022.
>
> [2] Jacot et al., "Neural Tangent Kernel: Convergence and Generalization in Neural Networks." In NeurIPS 2018.
>
> [3] Du et al., "Gradient Descent Provably Optimizes Over-parameterized Neural Networks." In ICLR 2019.
>
> [4] Chizat et al., "On the Global Convergence of Gradient Descent for Over-parameterized Models Using Optimal Transport." In NeurIPS 2018

---

> ### Author Response · Authors · 2025-11-28
>
> Dear Reviewer xcki,
>
> Thank you for your time and effort in reviewing our paper. As the discussion period is approaching its end, we wish to confirm whether our responses have adequately addressed your concerns. We would greatly appreciate it if you could re-evaluate our paper based on our rebuttal.
>
> Thank you again for your consideration and support.
>
> Best regards, Paper 15511 Authors

---

### Meta-Review · Area_Chair_1tB4 · 2026-01-19

**Summary:**

This paper studies weak-to-strong generalization and proposes a theoretical framework based on a generalized bias–variance decomposition for Bregman divergences. Reviewers generally acknowledge the technical effort and the intent to relax restrictive assumptions. However, overall support is lukewarm. A major concern, raised independently by multiple reviewers, is that the benefit of removing convexity assumptions is not demonstrated in a rigorous or convincing way: the key theoretical examples remain in convex regimes. Without a concrete non-convex example, it is unclear when the proposed theory can actually predict a non-zero performance improvement. As a result, several reviewers view the contribution as incremental relative to existing work.

In the rebuttal, the authors attribute the absence of non-convex examples to current limitations of deep learning theory. While this limitation is acknowledged transparently, it remains an important missing piece for a theory paper whose primary novelty is "non-convexity". The paper would be substantially strengthened by including even a highly simplified but analyzable non-convex instantiation.

After considering the rebuttal and discussion, I recommend rejection.

**Reviewer Concerns:**

* Reviewers MAkU, GSgt and WpDr all point out that while the paper emphasizes removing convexity assumptions, it does not provide a simple non-convex setting where the new theorem meaningfully strengthens what was previously known.

* Reviewers GSgt and WpDr found the original presentation unclear and difficult to follow. The rebuttal improves the writing and organization, which is explicitly acknowledged by Reviewer GSgt.

* Reviewer xcki raised concerns about the experimental scale and the apparent restriction of the analysis to linear models. These concerns are largely addressed in the rebuttal through experiments and clarifications.

**Reviewer Scores:**

While the rebuttal addresses several secondary concerns, the main issue remains unresolved: the paper does not rigorously demonstrate the benefit of removing the convexity assumption.

* Reviewer xcki is likely to raise the score to 8 since the rebuttal addresses all their concerns, though the confidence is 2.
* Reviewer MAkU may keep the original score 4.
* Reviewer GSgt is likely to raise the score to 4 for the improved presentation.
* Reviewer WpDr is likely to raise the score to 4 for the improved presentation.

---

### Decision · Program_Chairs · 2026-01-26

Reject